# Seeing to Generalize: How Visual Data Corrects Binding Shortcuts

**Nicolas Buzeta** [* 1 2] **Felipe del Rio** [1 2] **Cristian Hinostroza** [1 2] **Denis Parra** [1 2 3] **Hans Lobel** [1 2]
**Rodrigo Toro Icarte** [1 2]

## Abstract

Vision Language Models (VLMs) are designed to extend Large Language Models (LLMs) with visual capabilities, yet in this work we observe a surprising phenomenon: VLMs can outperform their underlying LLMs on purely text-only tasks, particularly in long-context information retrieval. To investigate this effect, we build a controlled synthetic retrieval task and find that a transformer trained only on text achieves perfect in-distribution accuracy but fails to generalize out-of-distribution (OOD), while subsequent training on an image-tokenized version of the same task nearly doubles text-only OOD performance. Mechanistic interpretability reveals that visual training changes the model's internal binding strategy: text-only training encourages positional shortcuts, whereas image-based training disrupts them through spatial translation invariance and other encoder-level inductive biases, forcing the model to adopt a more robust symbolic binding mechanism that persists even after text-only examples are reintroduced. We further characterize how binding strategies vary across training regimes and visual encoders, and show that analogous shifts occur during pre-trained LLM-to-VLM transitions. Our findings suggest that cross-modal training can enhance reasoning and generalization even for tasks grounded in a single modality.

## 1. Introduction

Large Language Models (LLMs) excel at a wide range of text-based tasks, from summarization to mathematical reasoning (Devlin et al., 2019; Brown et al., 2020; Wei et al., 2022; Ahn et al., 2024; Zhang et al., 2024b). However, on their own, they cannot leverage visual information. Vision–Language Models (VLMs) address this limitation by learning to encode images into token representations that a pre-trained LLM can consume to perform vision-centric tasks such as image captioning and visual question answering (Alayrac et al., 2022; Zhang et al., 2024a). A standard training pipeline begins by taking a pre-trained LLM and augmenting it with a vision encoder—typically a vision transformer (Dosovitskiy et al., 2021)—that maps visual inputs into a sequence of tokens (Li et al., 2023). These visual tokens are then fed into the LLM, and the combined system is trained end-to-end on tasks that require both visual and textual understanding (Bai et al., 2023; Liu et al., 2023).

In this paper, we study an intriguing observation: in some cases, VLMs outperform their underlying pre-trained LLMs on text-only tasks. That is, for certain tasks that require exclusively textual reasoning, the performance of the original LLM improves after being further trained on objectives that consider vision and language elements. We find that this scheme significantly and systematically enhances performance on information retrieval problems (Liu et al., 2024; Feng & Steinhardt, 2024; Gur-Arieh et al., 2026; Urrutia et al., 2026), especially in long-context settings. For instance, in our experiments, the VLM Qwen3-VL-8B (Yang et al., 2025) achieves 63.5% accuracy on a one-hop retrieval task, whereas its base model, Qwen3-8B, only reaches 48.2% on the same evaluation. This result is puzzling: why would training on image-based tasks lead to systematic improvements on purely text-based evaluations?

To study this question, we designed a set of controlled experiments centered on a text-only retrieval task (shown in Figure 1). At a high level, the task involves describing a set of colored shapes, assigning a letter to each shape, and querying the model for the letter associated with a shape of a specified color. For example, given the context description "*red circle and green triangle*" and the associations "*the circle is item_a and the triangle is item_b,*" the model is asked "*which item corresponds to the red shape*" (the correct answer being *item_a*). We trained a small transformer model to solve this task using contexts containing up to eight shapes (first row in Figure 1). The model quickly mastered the task,

---

[1] Department of Computer Science, Pontificia Universidad Católica, Santiago, Chile [2] Centro Nacional de Inteligencia Artificial (CENIA), Santiago, Chile [3] Instituto Milenio en Ingeniería e Inteligencia Artificial para la Salud (iHEALTH), Santiago, Chile. Correspondence to: Nicolas Buzeta <nicolas.buzeta@uc.cl>.

*Proceedings of the 43rd International Conference on Machine Learning*, Seoul, South Korea. PMLR 306, 2026. Copyright 2026 by the author(s).

achieving perfect in-distribution accuracy—i.e., when the input contained at most eight objects. However, its out-of-distribution (OOD) generalization was poor: on contexts with more than eight shapes, its accuracy dropped to 37.2%.

We then continued training the same transformer on an equivalent retrieval task, replacing the textual context descriptions with tokenized images of the shapes and their colors (second and third rows in Figure 1). Training initially relied exclusively on image-based contexts, and we later introduced a mixture of image-only and text-only instances. Crucially, throughout all stages of training, the model was exposed only to contexts containing at most eight objects. After this additional training, the model exhibited the same puzzling phenomenon observed in pre-trained models: the vision-language model showed a substantial OOD performance improvement over the base model on the original text-only retrieval task, rising from 37.2% to 69.5%.

Having replicated the phenomenon in a controlled setting, we use mechanistic interpretability to examine how visual training alters the language model's internal computations. We find that visual training fundamentally changes the model's binding strategy—the mechanism by which it links information across tokens. Prior work has shown that LLMs employ different binding mechanisms to solve in-context reasoning problems (e.g., Gur-Arieh et al., 2026; Urrutia et al., 2026). In particular, positional binding relies on token positions within the sequence, whereas symbolic binding uses the semantic content of tokens to establish associations. Following the methodology of Gur-Arieh et al. (2026), we find that a model trained only on text almost exclusively relies on positional binding. In our retrieval task, this manifests as a shortcut (Geirhos et al., 2020): the model exploits positional regularities in contexts containing up to 8 shapes, instead of attending to semantic content.

However, once the model is further trained on image-based versions of the task, positional binding becomes ineffective. Unlike sequences, images lack a canonical ordering, and visual models are typically designed to be robust to spatial translations. Consequently, spatial layouts do not reliably correspond to token ordering, rendering position-based shortcuts unreliable. This disruption—driven partially by translation invariance, though likely compounded by other encoder-level inductive biases—pushes the model toward a more robust strategy and predominantly symbolic binding. Remarkably, when we subsequently reintroduce text-only examples, the model continues to use this new, more robust symbolic binding strategy to solve the original task. This shift in binding behavior explains the improved OOD performance: symbolic binding is inherently better suited for information retrieval, particularly in settings involving long or variable-length contexts.

We conclude the paper with a detailed analysis of the differ-

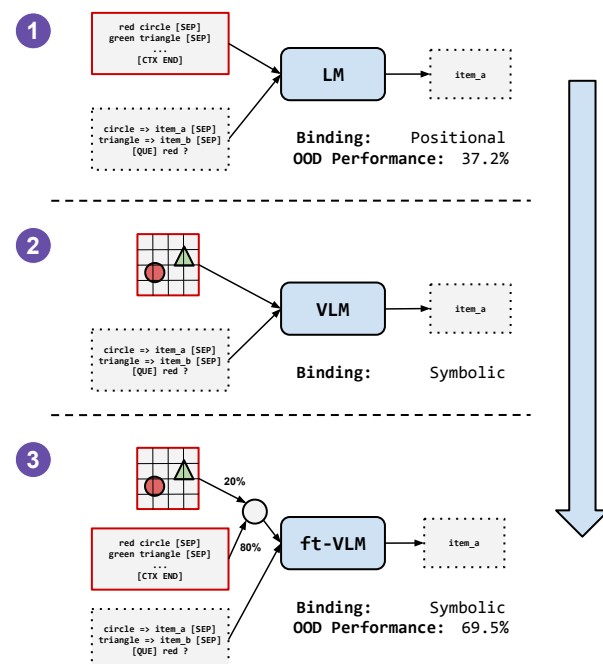

*Figure 1.* Overview of the training pipeline, showing the progression from text-only training to vision–language training.

ent binding mechanisms that emerge under various training regimes. While incorporating images consistently shifts the model toward symbolic binding, we find that the specific strategy used to solve the task appears to be contingent on the choice of visual encoder and can vary significantly even across different random initializations. We further include sensitivity analyses showing that the observed performance gains cannot be attributed to indirect exposure to longer contexts during training. Finally, through experiments on pre-trained models, we confirm that transitioning from an LLM to its VLM variant indeed makes the internal binding mechanism more symbolic and less positional. We believe these findings open new avenues for understanding how cross-modal training can enhance reasoning and generalization—even on tasks that rely solely on another modality.

## 2. VLM Gains on Text-Only Tasks

We first show that VLMs outperform their base LLM on text-only tasks that require retrieving information from long contexts. To study this effect, we evaluate four pre-trained models: Qwen 2, 2.5, and 3 (Yang et al., 2024a;b; 2025), as well as InternLM 3 (Team, 2025). All models were evaluated as released, without any task-specific fine-tuning or post-processing (see Appendix B for further details).

To evaluate retrieval abilities, we employ two synthetic tasks: *Direct Retrieval* and *Indirect Retrieval*. In the Direct Retrieval task, the objective is to determine where a person

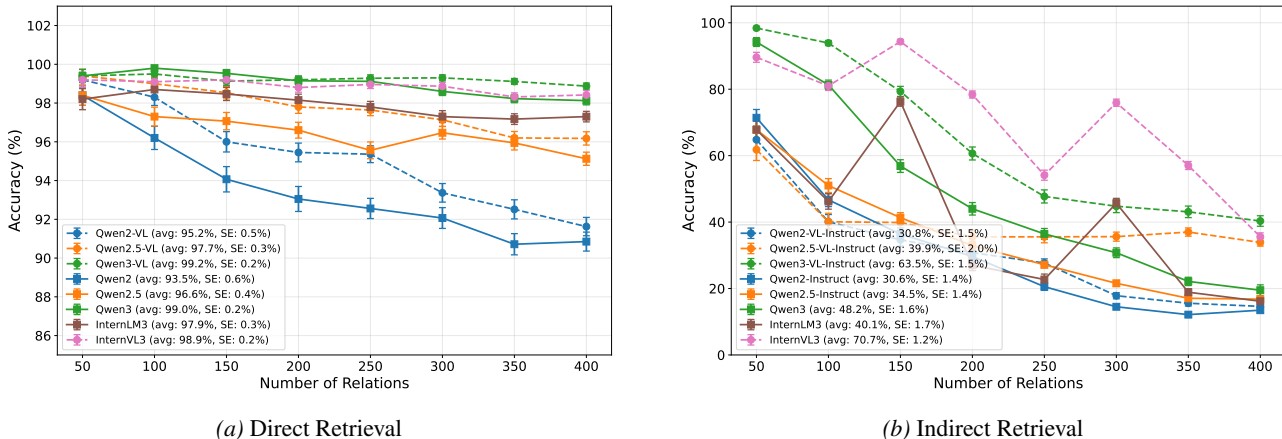

*(a)* Direct Retrieval  *(b)* Indirect Retrieval

*Figure 2.* Binding accuracy comparison between text-only LLMs and their VLM counterparts across four models (Qwen 2, 2.5, 3 and InternLM 3). VLMs consistently outperform LLMs as context length increases. The left panel shows performance on the simpler direct retrieval task, while the right panel shows the harder indirect retrieval task. Individual breakdown per model is provided in Appendix A.

lives given a large textual context. Formally, let $\mathcal{N}$ denote a set of names and $C$ a set of cities. Each input consists of a context containing $l$ statements of the form "$n_i$ lives in $c_i$," where each $n_i \in \mathcal{N}$ and each $c_i \in C$ appears exactly once within the context. The query then asks for the city associated with one of the names mentioned in the context. For example (with the model's output highlighted in blue):

> **Context:** James lives in Tokyo. Mary lives in Paris. John lives in Mumbai. **Question:** In what city does John live? John lives in Mumbai

The Indirect Retrieval task introduces an additional reasoning step. In addition to $\mathcal{N}$ and $C$, we define a set of foods $\mathcal{F}$. The context is divided into two parts. The first contains statements of the same form as in the Direct Retrieval task, "$n_i$ lives in $c_i$". The second contains $l$ statements of the form "$f_i$ is liked by $n_i$." Each name $n_i$ appears exactly once in each part to avoid ambiguity, so the index $i$ links a food $f_i$ to a city $c_i$ through the same person $n_i$. The query refers to a food rather than a name, requiring the model to first identify the corresponding person and then retrieve their city. For instance, a concrete example with three pairs (with the model's output highlighted in blue) would be:

> **Context:** Banana is liked by Mary. Orange is liked by John. Mary lives in Paris. John lives in Mumbai. **Question:** Where does the person that likes Orange live? Answer with just the city name. **City:** Mumbai

Both tasks are inspired by the *Capitals* task of Feng & Steinhardt (2024). Direct Retrieval task tests basic context lookup, while Indirect Retrieval task adds a compositional retrieval step. Importantly, neither task requires prior world knowledge (e.g., real-world capitals), since all necessary information is contained within the provided context.

Figure 2 presents the results for both the Direct and Indirect

Retrieval tasks across the Qwen and InternLM families. For each context length $l$, we generated 10 independent contexts. Each model was evaluated on every query position within each context, resulting in a total of $10 \times l$ queries for that length. We report the mean accuracy and standard error for each model across all evaluated context lengths.

In the Direct Retrieval task, VLMs consistently outperform their LLM counterparts across nearly all context lengths, with Qwen3-VL models achieving near-ceiling accuracies of approximately $98\%$ even at $l = 400$. This performance gap becomes even more pronounced in the Indirect Retrieval task, where VLMs demonstrate a clear and persistent advantage over their corresponding base LLMs. This outcome is particularly surprising given that VLMs are trained primarily to enable LLMs to interpret visual information. Why, then, do VLMs also exhibit consistently superior performance on purely text-based retrieval tasks?

In the following section, we aim to reproduce this phenomenon in a controlled setting, which will allow us to investigate it using mechanistic interpretability tools.

## 3. Problem Formulation

To isolate the role of modality in the performance gap between VLMs and LLMs, we introduce a modified Indirect Retrieval task that enables controlled switching between modalities while keeping the underlying task unchanged.

The task links colored shapes and unique target labels. The model must first track objects defined by both their shape and color (e.g., a red triangle). It is then presented with a separate list of associations (e.g., "the triangle is item a"). The challenge is to retrieve the correct target item when queried using only the object's color (e.g., returning item a when asked about "red"), as illustrated in the first step of Figure 1.

Using simple visual primitives such as shapes and colors allows the task can be presented identically across modalities—either as a textual description or as an image—without altering its underlying structure (see Figure 1).

We now formally define our Indirect Retrieval task. Let $\mathcal{A}$ be a set of attributes (e.g., colors), $\mathcal{E}$ a set of entities (e.g., shapes), and $\mathcal{I}$ a set of items (e.g., item001). The goal is to map a query attribute $q \in \mathcal{A}$ to a target item $y \in \mathcal{I}$ through an intermediate entity $e \in \mathcal{E}$. Solving the task requires two reasoning steps. First, the model must identify the entity $e$ associated with the query attribute $q$ using the context set $\mathcal{C} = (a_i, e_i)_i$, where $a_i \in \mathcal{A}$ and $e_i \in \mathcal{E}$. Second, it must retrieve the item $y$ linked to that entity using the association set $\mathcal{B} = (e_j, y_j)_j$, where $e_j \in \mathcal{E}$ and $y_j \in \mathcal{I}$. We consider two instantiations of this task: one presented entirely in text and another that combines images with text.

The unified prompt structure for both modalities is:

$$\mathbf{x} = [\mathbf{X}_{\text{context}}, [\texttt{CTX\_END}], \mathbf{X}_{\text{associations}}, [\texttt{QUE}], \mathbf{x}_{\text{query}}]$$

where $\mathbf{X}_{\text{associations}}$ is a sequence of text tokens defining the Entity $\rightarrow$ Item assignments (e.g., shape01$\rightarrow$item01), while [CTX\_END] and [QUE] are special delimiter tokens. Then, for both tasks, we define the context $\mathbf{X}_{\text{context}}$ as a sequence of attribute-entity pairs $(a_i, e_i)$.

In the **text modality**, the context is represented as follows:

$$\mathbf{X}_{\text{context}}^{\text{text}} = [a_1, e_1, a_2, e_2, \ldots, a_N, e_N]$$

where each pair is described sequentially (e.g., "red square"). In the **vision modality**, the context is a sequence of image patches where each entity $e_i$ is rendered with attribute $a_i$:

$$\mathbf{X}_{\text{context}}^{\text{image}} = [\texttt{}_1, \texttt{}_2, \ldots, \texttt{}_N]$$

## 4. VLM Gains in a Controlled Setting

In this section, we replicate the performance gains of VLMs over LLMs reported in Section 2 by training a 12-layer decoder-only Transformer on the Indirect Retrieval task. Exact prompt structures are provided in Appendix C, and full model and optimization details in Appendix D. To accomplish this, we first train the model exclusively on the text modality, where the context $\mathbf{X}_{\text{context}}^{\text{text}}$ consists of sequential token pairs representing attribute–entity associations (e.g., "red triangle", "blue circle"). Training follows a curriculum that gradually increases task complexity: we begin with contexts containing 2 object pairs and a restricted vocabulary, then progressively expand vocabulary and context length until contexts contain up to 8 objects and validation performance saturates, resulting in the baseline text-only model $\mathcal{M}_{\text{text-only}}$ (see Appendix D for the full curriculum details).

Next, we continue training the same model exclusively on the image modality. As described in Section 3, the Indirect Retrieval task can be instantiated in both text and vision

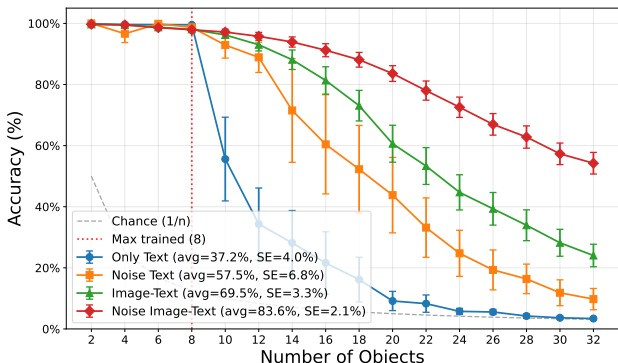

*Figure 3.* Combined generalization performance across all models. **Text-Only** (Blue) suffers from sharp degradation on OOD lengths (37.2% average). **Image-Text** (Green) significantly improves generalization (69.5% average). **Noise-Text** (Orange) benefits from positional range expansion but remains insufficient (57.5% average). **Noise-Image-Text** (Red) achieves the strongest robustness (83.6% average). Individual plots are provided in Appendix E.

modalities. In the vision setting, the context $\mathbf{X}_{\text{context}}^{\text{image}}$ consists of rendered images in which each entity is depicted with its associated attribute (e.g., colored shapes). To study the impact of visual training, we replace the text context with token representations extracted from a frozen, pre-trained image encoder. We experiment with three encoder architectures: a supervised CNN-based ResNet-152 (He et al., 2016), a supervised Transformer-based ViT-B/16 (Dosovitskiy et al., 2021), and a self-supervised Transformer-based DINOv3 (Siméoni et al., 2025). Each variant is trained until validation performance saturates.

Finally, we transfer the model back to the text domain while keeping the restricted parameter set (8 colors, 13 shapes, 8 items) and train on a mixed curriculum comprising 20% image-based and 80% text-based tasks. For OOD evaluation, we expand the training distribution to match the full complexity of the baseline text model (216 colors, 216 shapes, 32 items), yielding the final model $\mathcal{M}_{\text{image-text}}$.

Figure 3 compares the two models on the Indirect Retrieval task in the text modality as the number of objects in the context increases. We train $\mathcal{M}_{\text{text-only}}$ with four random seeds. From each, we initialize and train separate image-modality variants using three image encoders (ResNet-152, ViT, and DINOv3), resulting in 12 runs [1] for $\mathcal{M}_{\text{image-text}}$.

As the figure shows, the text-only model, $\mathcal{M}_{\text{text-only}}$ (blue curve), generalizes poorly to OOD sequence lengths: its accuracy drops sharply beyond the training maximum of eight items, yielding an average OOD accuracy of 37.2%.

---

[1]However, we discarded one run that failed to learn the text task beyond random performance, to measure generalization only on models that learned the task in-distribution (see Appendix D.8).

In contrast, $\mathcal{M}_{\text{image-text}}$ (green curve) shows substantially stronger generalization, achieving an average OOD accuracy of 69.5%. This large gain demonstrates that training on visual inputs enhances generalization in the text modality, mirroring the phenomenon observed in large-scale VLMs.

**Noise Augmentation.** One possible hypothesis about why $\mathcal{M}_{\text{image-text}}$ outperforms $\mathcal{M}_{\text{text-only}}$ is that the improvement arises from exposure to longer sequences during training. Image encoders produce sequences of patch tokens (e.g., 196 tokens for a $14 \times 14$ grid), meaning the model encounters a broader range of positional indices than in text-only training, where contexts contain at most 8 object pairs.

To test whether exposure to longer contexts alone explains the performance gap, we fine-tune both $\mathcal{M}_{\text{text-only}}$ and $\mathcal{M}_{\text{image-text}}$ with unattendable noise tokens (Shen et al., 2023) inserted between context pairs. We denote the resulting models as $\mathcal{M}_{\text{noise-text}}$ and $\mathcal{M}_{\text{noise-image-text}}$, respectively.

The results in Figure 3 show that adding noise tokens does improve the OOD performance of the text-only model, but this effect is insufficient to account for the full gains observed with visual training. Instead, the visual modality appears to provide an additional form of regularization beyond what is achievable through simple context-length expansion.

# 5. Explaining the Gains: Image Training Induces Binding Strategy Shifts

The previous section established that visual training substantially improves OOD generalization beyond what can be explained by increased exposure to longer positional ranges. In this section, we show that the transition from $\mathcal{M}_{\text{text-only}}$ to $\mathcal{M}_{\text{image-text}}$ corresponds to a shift in how the model binds and retrieves information. Specifically, the dominant binding strategy moves from a brittle, position-dependent mechanism to a more robust, symbolic one.

We first demonstrate this shift in the clean setting, without noise augmentation. We then show that noise injection induces an intermediate mixed binding strategy in the text-only model. Finally, we connect these controlled findings to large pre-trained models, showing that VLMs also rely more heavily on symbolic binding than their LLM counterparts.

## 5.1. Background: Binding Mechanisms in Transformers

Recent work has identified multiple mechanisms that transformer models use for variable binding. Gur-Arieh et al. (2026) describe three main strategies. In the **positional** mechanism, the model retrieves an entity based on its ordinal position in the sequence (e.g., "the second entity"), relying heavily on positional encodings and making it sensitive to sequence structure. In the **symbolic mechanism**, retrieval is content-based: the query acts as a key that matches the associated entity in a position-agnostic way. This mechanism is referred to as *lexical* in Gur-Arieh et al. (2026) and *symbolic* in prior work on variable binding (Wu et al., 2025); we adopt the term **symbolic** because it extends naturally to visual inputs, where "lexical" matching is less well-defined. Finally, the **reflexive** mechanism relies on pointer-like references and is needed when the target appears earlier than the query, preventing direct attention from query to target.

In our task, the attribute (query) consistently precedes the entity (target), making symbolic binding viable and reducing the importance of the reflexive mechanism. We therefore focus our analysis on the competition between positional and symbolic strategies.

## 5.2. Methodology: Identifying Binding Mechanisms

To determine which binding mechanism our models rely on, we use the interchange intervention proposed by Gur-Arieh et al. (2026). This method enables causal identification of binding strategies by constructing paired inputs—an original example and a counterfactual—designed so that different mechanisms would produce different predictions.

The key idea is to create input pairs where a **positional** strategy would retrieve the entity solely based on its position in the sequence, which changes between the original and counterfactual, while a **symbolic** strategy would instead retrieve the entity associated with the query's semantic content, which may remain unchanged across the pair. By patching activations from the counterfactual run into the original run at different layers, we can measure which mechanism dominates the model's computation. Full methodological details are provided in Appendix F.

## 5.3. The Clean Case: A Complete Mechanism Shift

We begin by analyzing the binding mechanisms without noise augmentation and observe a clear dichotomy between the text-only and image-trained models. Figure 4 reports, for each layer, the proportion of times the model's binding mechanism is classified as positional, symbolic, reflexive, or mixed when solving text-modality inputs containing up to eight objects. The results show that the dominant binding mechanism remains largely consistent across layers, except for the final layer. In this last layer, $\mathcal{M}_{\text{text-only}}$ adopts an almost exclusively positional strategy, meaning it heavily relies on token positions to produce the correct answer. In contrast, $\mathcal{M}_{\text{image-text}}$ primarily uses a symbolic binding mechanism, matching semantic content directly rather than depending on positional indices.

A symbolic mechanism naturally generalizes to longer contexts: because retrieval is based on semantic matching, the strategy remains valid regardless of sequence length. Positional binding, however, relies on counting or locating "the $i$-

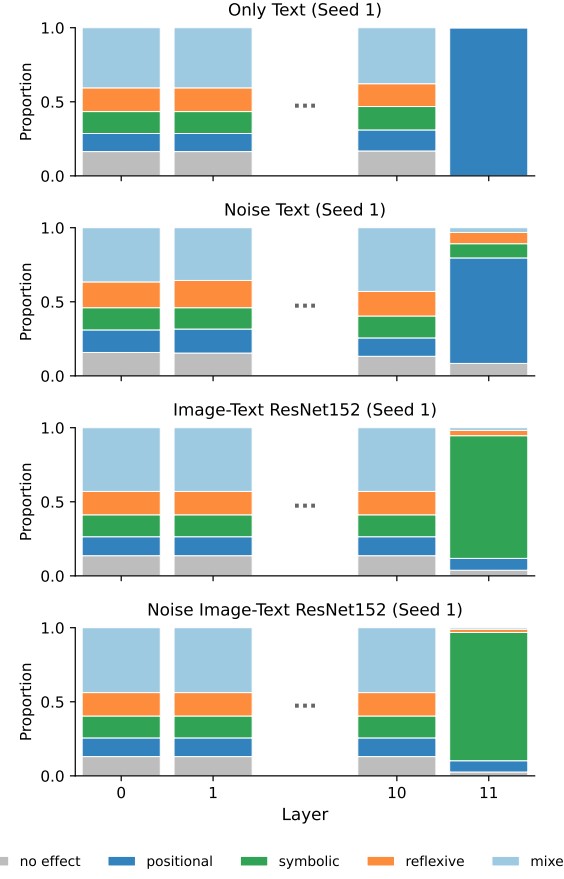

*Figure 4.* **Binding Mechanism Shift.** Interchange intervention results showing the dominant binding mechanism at each layer across all model variants. Text-only models ($\mathcal{M}_{\text{text-only}}$) rely on the **positional** mechanism (blue). Noise augmentation ($\mathcal{M}_{\text{noise-text}}$) introduces an increase in symbolic binding, but the model remains predominantly positional. Image-trained models ($\mathcal{M}_{\text{image-text}}$ and $\mathcal{M}_{\text{noise-image-text}}$) both transition to a **symbolic** mechanism (orange).

th binding" within the sequence, which tends to break when the model is asked to retrieve items in contexts longer than those seen during training. This distinction explains why $\mathcal{M}_{\text{image-text}}$ generalizes better than $\mathcal{M}_{\text{text-only}}$ in the OOD evaluation presented in Section 4.

### 5.4. Why Images Induce the Mechanism Shift

A key difference between the two modalities is **translation invariance**, which is intrinsic to models trained on visual inputs but absent from sequential text. We hypothesize that this property plays a central role in driving the shift in binding strategy.

In the text modality, object pairs appear at specific, fixed positions in the sequence. A model can exploit this structure by learning to associate positions rather than content. This positional shortcut is highly effective within the training distribution, as tracking fixed positions is a simpler, more

direct mechanism than developing a robust symbolic binding strategy that actively matches semantic features.

In the image modality, however, object positions are sampled uniformly at random across the image canvas, with no fixed spatial regions. The same attribute-entity pair (e.g., a red circle) can appear at any location in the image, and this location carries no task-relevant information. Positional binding is therefore ineffective at solving the visual task.

This creates pressure for the model to discover an alternative: symbolic binding. By learning to match attributes to entities based on semantic identity rather than position, the model can solve the visual task regardless of where objects appear. When the model is subsequently transferred back to text, it retains this symbolic strategy, which proves more robust to distribution shifts than the positional approach it would have learned from text alone.

To test this hypothesis, we conducted an additional experiment in which object positions were fixed across all images, thereby reducing spatial variability at the task level (while the image encoders remain inherently translation-invariant). Surprisingly, the results were mixed: across twelve random seeds, five models retained predominantly positional binding after visual training, consistent with our hypothesis that translation invariance drives the shift from positional to symbolic binding. However, in the remaining seven seeds, the models still transitioned to symbolic binding despite the lack of spatial variation.

One possible explanation is that the pre-trained image encoder introduces multiple inductive biases—beyond translation invariance—arising from its architecture and pretraining objective. Consequently, even when explicit spatial variation is minimized, these biases may still disrupt positional shortcuts learned during text-only training. Overall, our results confirm that training on images shifts the core binding mechanism and that translation invariance plays a central role in this effect. However, we cannot rule out the influence of additional factors (see Appendix G for details).

### 5.5. The Noise Case: Mixed Mechanisms

We now consider the case where noise tokens are added. The results are shown in Figure 4. As the figure illustrates, $\mathcal{M}_{\text{noise-text}}$ exhibits a slightly different profile from the pure text model: its binding strategy remains predominantly positional, but with a modest increase in symbolic behavior. Introducing unattendable noise tokens disrupts the clean positional regularities present in the synthetic text data, making strict positional counting less reliable and encouraging the model to rely partially on semantic matching.

This controlled effect mirrors observations in large pre-trained LLMs. Prior work shows that different families of models use a mixture of positional and symbolic mech-

anisms rather than purely positional strategies (Gur-Arieh et al., 2026). Our results suggest a possible explanation: natural language is inherently irregular and less positionally structured than our synthetic task, introducing variability that weakens positional shortcuts and promotes partial symbolic binding. We therefore hypothesize that exposure to unstructured text during pre-training nudges models toward mixed mechanisms.

However, this shift is incomplete. Despite its increased symbolic component, $\mathcal{M}_{\text{noise-text}}$ (57.5% OOD) still substantially underperforms $\mathcal{M}_{\text{image-text}}$ (69.5% OOD), indicating that noise alone does not induce fully robust binding. Visual training appears to exert a stronger and more direct pressure toward symbolic strategies, changing which mechanism is viable rather than merely degrading positional cues.

Consistent with this interpretation, $\mathcal{M}_{\text{noise-image-text}}$ combines both effects: it exhibits symbolic binding similar to $\mathcal{M}_{\text{image-text}}$ yet achieves even higher OOD generalization (83.6%). This suggests complementary roles: visual training drives the mechanism shift, and noise exposure further improves generalization, likely through broader positional coverage without altering the dominant binding strategy.

### 5.6. Validation on Large-Scale Models

To assess whether our findings extend beyond controlled settings, we performed interchange interventions on the Qwen (2, 2.5, and 3) and InternLM (3) model families, using the same code and task setup as Gur-Arieh et al. (2026). Our goal is to test the hypothesis that image training shifts the core binding mechanism in large-scale models from positional to symbolic, which may help explain the superior performance of VLMs over their LLM counterparts on long-context retrieval tasks.

Figure 5 reports the symbolic-to-positional binding ratio per layer across models, defined as the frequency of symbolic binding relative to positional binding during retrieval. A ratio greater than 1 indicates a preference for symbolic binding. As shown, all VLMs exhibit higher ratios on average than their LLM counterparts, indicating a stronger reliance on symbolic representations.

The main exception is Qwen 2, where the LLM shows a higher peak ratio than the VLM; however, the VLM remains more symbolic on average (1.61 vs. 1.48). Notably, Qwen 2 also exhibits the smallest performance gap between LLM and VLM in retrieval tasks (Figure 2). In contrast, InternLM 3 shows the largest improvement in retrieval performance, which coincides with the largest increase in the symbolic-to-positional ratio (+0.37) from LLM to VLM.

Overall, these results reinforce the hypothesis that the improved retrieval performance of VLMs is linked to a shift in binding mechanism, and that incorporating image data

during training plays a key role in driving this transition, although other contributing factors cannot be ruled out.

In the next section, we move from behavior to circuit-level analysis, dissecting the attention patterns and information flow that implement positional and symbolic binding.

## 6. Characterizing the Binding Circuits

Having established *which* binding mechanisms each model employs, we now analyze *how* these mechanisms are implemented at the circuit level. Using attention knockout experiments (Geva et al., 2023) to identify critical components and linear probes to decode information flow, we uncover three circuit architectures: the Positional Circuit (text-only models) and two variants of the Symbolic Circuit (image-trained models). Figure 6 illustrates the information flow in each circuit type.

### 6.1. The Three Circuits

**The Positional Circuit.** The circuit operates in two independent streams: (1) the color query token identifies the position of its matching attribute in the context, and (2) the association shape token independently finds its matching shape and stores that context position. These streams never exchange attribute information—the "binding" is implicit, relying solely on shared position indices. The answer token retrieves the correct item by matching position indices, effectively counting to the correct output.

**Symbolic Circuit A (Color-Key).** This variant uses color as the retrieval key. The circuit actively moves content: context color is first copied into the context shape token, creating an explicit attribute-entity bundle. Then, the association shape retrieves this stored color from the context. Finally, both the association item and answer token converge on this color information to produce the correct output.

**Symbolic Circuit B (Shape-Key).** Similar to Circuit A, the context color is initially copied into the context shape. However, the query color looks back at the context to find the shape token marked with its color, copying the shape context into its activations. The answer token uses this shape information for retrieval.

Despite their different intermediate keys, both symbolic variants share the fundamental property of transferring semantic identity rather than positional indices. This explains the robustness to sequence length observed in our results.

### 6.2. Validating the Circuits

We validate these circuit architectures using attention knockouts to identify critical pathways and linear probes to decode information flow at each position (see Appendix H for detailed analysis). The results confirm our predictions:

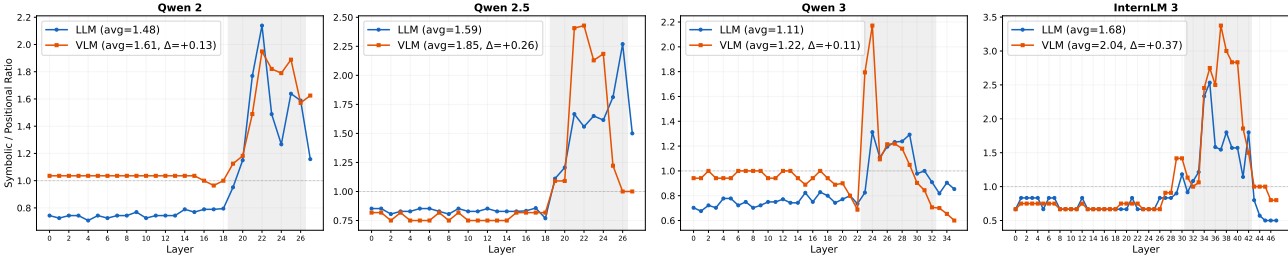

*Figure 5.* **Per-layer symbolic-to-positional ratio.** Each panel shows the ratio across layers for a model pair. VLM variants (orange) consistently exhibit higher ratios than their LLM counterparts (blue), though the peak layer varies.

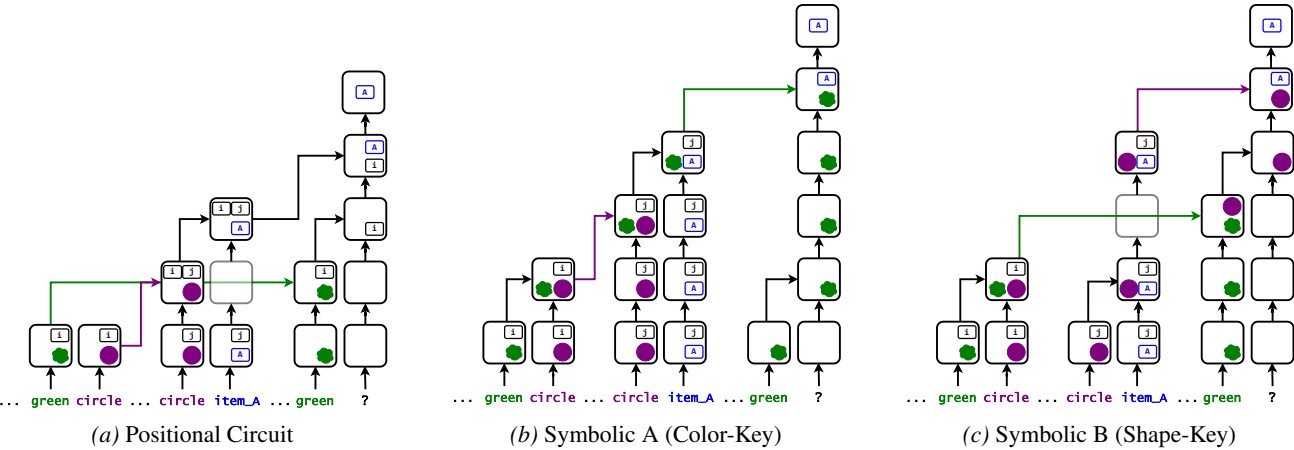

*(a)* Positional Circuit         *(b)* Symbolic A (Color-Key)         *(c)* Symbolic B (Shape-Key)

*Figure 6.* **Circuits corresponding to the binding mechanisms.** Early layers encode token-level information (color, shape, item identity). Colored arrows indicate attention-mediated information transfer between tokens. The key distinction: the Positional Circuit transfers only position indices (implicit binding), while Symbolic Circuits transfer semantic content (explicit binding).

**Positional Circuit.** Knockouts reveal critical pathways between tokens involved in position-matching. Probes show high position decodability, but crucially *low* attribute decodability at entity positions—the model never explicitly represents "green circle" as a bound unit.

**Symbolic Circuits.** Knockouts identify the attention pathways responsible for transferring attribute information. Probes reveal the **binding signature**: a marked surge in attribute decodability at entity positions as attributes are explicitly bound. This signature distinguishes symbolic from positional binding.

### 6.3. Extension to Large-Scale Models

We extended our probing analysis to the Qwen model family (Qwen 2, 2.5, and 3). The VLM variants exhibit the same binding signature observed in our small-scale experiments: an early rise in attribute decodability at the context entity token, followed by a rise at the binding entity token. This pattern is consistent with Symbolic Mechanism A. Text-only baselines show significantly lower attribute decodability at these positions, confirming their reliance on positional heuristics (see Appendix H for further details).

## 7. Related Work

**Multi-modal training effects on language models.** Recent work has systematically compared multi-modal training regimes against their language-only backbones to understand how visual grounding reshapes linguistic capabilities. From a theoretical standpoint, multi-modal training has been argued to promote robustness and generalization by discouraging reliance on modality-specific correlations (Xue et al., 2024) and by inducing a more faithful and structured latent semantic space (Huang et al., 2021). Empirically, some studies show that VLMs can match or even improve text-only performance, with Dai et al. (2024b) reporting gains in language understanding, mathematics, coding, and reasoning, and Ratzlaff et al. (2025) observing improvements in commonsense reasoning. However, these benefits are not uniform: multi-modal training can also introduce regressions, such as degraded mathematical reasoning in some LLaVA-style models (Ratzlaff et al., 2025).

**Mechanistic Interpretability.** In this work, we adopt a mechanistic interpretability perspective to analyze the internal computations of Transformers. We combine interchange interventions (Meng et al., 2022; Geiger et al., 2021; Fin-

| MODEL | ARC-C | ARC-E | BOOLQ | HS | MMLU | OBQA | PIQA | WG | MEAN |
|---|---|---|---|---|---|---|---|---|---|
| QWEN 3 | 56.23 | 80.93 | 86.67 | 74.9 | 73.05 | 41.8 | 77.53 | 68.11 | **69.9** |
| QWEN 3 VL | 46.5 | 69.94 | 87.34 | 54.91 | 65.95 | 41.8 | 75.03 | 59.43 | 62.6 |
| QWEN 2.5 | 55.12 | 81.27 | 86.42 | 80.45 | 71.69 | 49 | 80.25 | 70.8 | **71.9** |
| QWEN 2.5 VL | 45.73 | 64.81 | 84.95 | 70.09 | 66.83 | 44 | 75.3 | 65.59 | 64.7 |
| QWEN 2 | 53.8 | 76.64 | 85.44 | 80.61 | 69.94 | 46.4 | 80.41 | 69.53 | **70.3** |
| QWEN 2 VL | 48.72 | 70.62 | 86.85 | 74.87 | 67.4 | 47.6 | 78.84 | 67.8 | 67.8 |

*Table 1.* Comparison of VLMs and LLMs across eight standard text benchmarks.

layson et al., 2021; Vig et al., 2020; Geiger et al., 2020) to causally identify which hidden activations drive specific outputs and how binding and compositional information is propagated (Feng & Steinhardt, 2024; Saravanan et al., 2025; Gur-Arieh et al., 2026; Wu et al., 2025), attention knockout techniques (Geva et al., 2023; Gur-Arieh et al., 2026) to assess the functional role of specific attention pathways, and linear probing methods (Alain & Bengio, 2017; Ravichander et al., 2021; Belinkov, 2022) to characterize which variables are explicitly encoded across layers.

**Binding.** Binding—the ability to correctly associate entities with their attributes (Treisman, 1996; Feng & Steinhardt, 2024)—has recently been examined through mechanistic analyses that probe how Transformers implement and represent such associations. Prior work has leveraged interchange interventions (Feng & Steinhardt, 2024; Saravanan et al., 2025; Gur-Arieh et al., 2026; Prakash et al., 2026; Wu et al., 2025) and analyses of attention head behavior (Wu et al., 2025; Urrutia et al., 2026) to characterize how binding information is encoded, propagated, and learned within the network (Wu et al., 2025). We build directly on this line of research, grounding our analysis in the taxonomy of binding mechanisms introduced by Gur-Arieh et al. (2026), which distinguishes between positional (Dai et al., 2024a; Prakash et al., 2024; 2026), symbolic, and reflexive binding, and use it as a unifying framework to interpret the binding behaviors observed in our models.

## 8. Conclusion

Our work revealed an intriguing phenomenon: VLMs consistently outperformed LLMs on information retrieval tasks, even when the inputs were purely textual. We replicated this effect in a controlled setting and found that training on images induced a fundamental shift in the binding mechanism—from positional to symbolic. This shift provided a compelling explanation for the superior retrieval performance of VLMs despite the tasks being text-only.

Our findings highlight cross-modal training as a powerful form of inductive bias: even when evaluation is unimodal, exposure to another modality can promote more robust internal computations. More broadly, this suggests that mul-

timodal objectives may offer a principled way to mitigate shortcut learning and strengthen generalization.

At the same time, it is important to note that the benefits of multimodal training appear task-dependent. As shown in Table 1, VLMs generally underperform their LLM counterparts on standard text benchmarks. This is intuitive, as multimodal training may affect language-only performance through several mechanisms. For instance, model capacity may be shared across modalities, reducing specialization for purely textual tasks, or multimodal fine-tuning may introduce interference with representations that were useful for language understanding and world knowledge. Overall, understanding when and why multimodal training improves unimodal reasoning remains an open and interesting question. Future work should clarify which task properties benefit from cross-modal inductive biases.

## 9. Limitations and Future Work

While we identify translation invariance as a key driver of the binding shift (Section 5), the precise mechanisms remain to be fully understood. In particular, why some training runs transition to symbolic binding even with fixed object positions is an open question.

More broadly, it remains unclear whether other modalities, such as audio or video, would induce similar shifts in binding strategies. Multimodal training may improve generalization by introducing more diverse learning signals and inductive biases, potentially enhancing out-of-distribution robustness (del Rio et al., 2025). Extending these effects beyond vision is an important direction for future work.

Finally, improved retrieval performance is a key capability that should not be underestimated. Most reasoning and in-context learning tasks fundamentally rely on retrieving relevant information from the context. In this sense, effective retrieval is a necessary—though not sufficient—condition for solving such tasks. Understanding how to leverage the improved retrieval capabilities of VLMs for downstream reasoning remains an important direction for future work.

**Reproducibility.** Code is available at `https://github.com/nicobuzeta/seeing-to-generalize`.

## Acknowledgements

We gratefully acknowledge the support of the Agencia Nacional de Investigación y Desarrollo (ANID), Chile. This research was funded by the National Center for Artificial Intelligence CENIA FB210017, Basal ANID. The work of N. Buzeta was supported by ANID National Master's Scholarship (Beca de Magíster Nacional). The work of D. Parra was supported by Fondecyt Regular 1231724 and the Millennium Initiative research centers iHealth ICN2021_004 and IMFD ICN17_002. The work of R. Toro Icarte was supported by Fondecyt Iniciación 11230762. The work of H. Lobel was supported by Fondecyt Regular 1241772.

## Impact Statement

This paper presents work whose goal is to advance the field of machine learning. There are many potential societal consequences of our work, none of which we feel must be specifically highlighted here.

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

# A. Detailed Performance Breakdown

Figure 7 presents the full performance breakdown for all evaluated model families. The first two rows show Qwen model generations (Direct and Indirect Retrieval), and the third row shows the InternLM 3 family. Across all families, the VLM variants maintain higher accuracy at longer context lengths compared to their text-only counterparts.

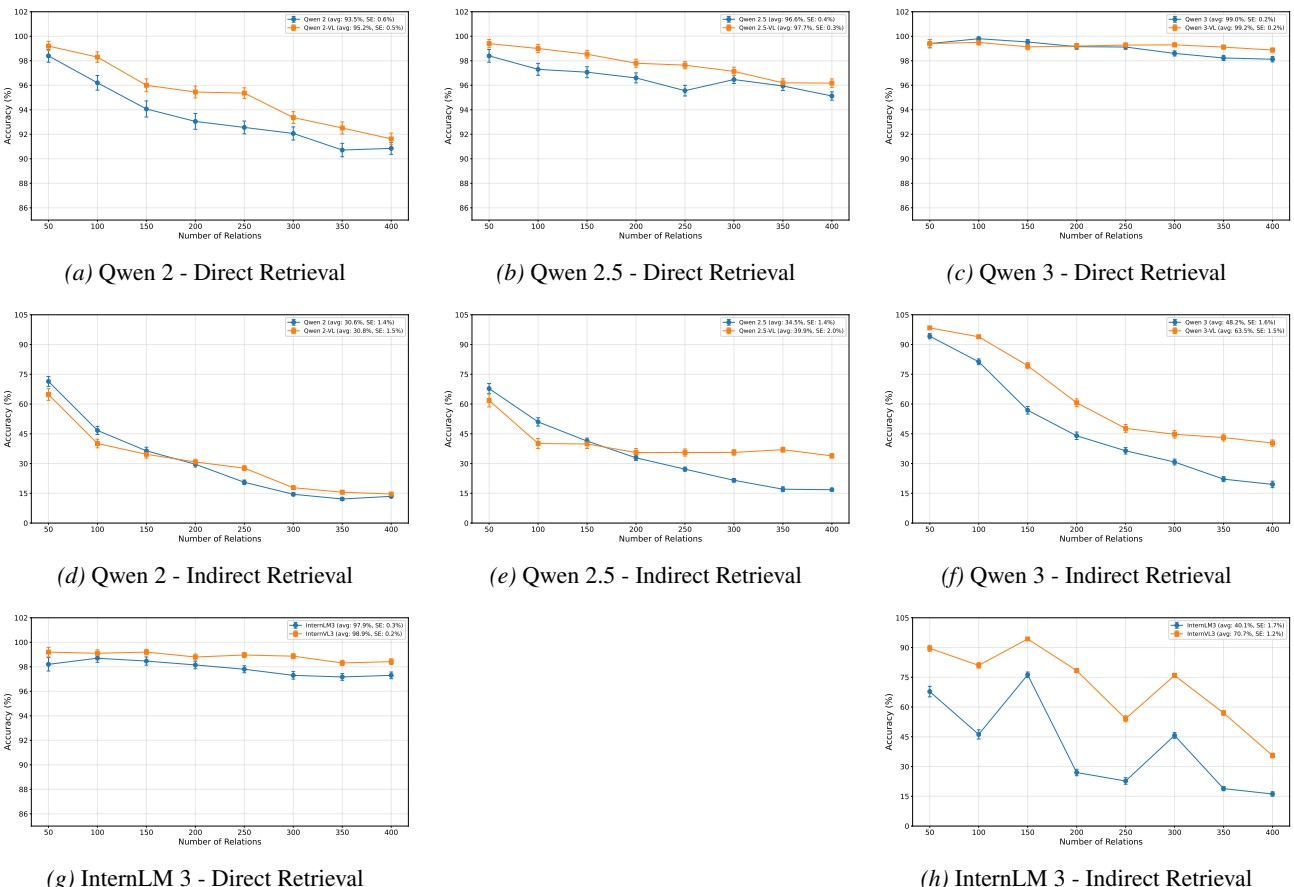

*(a)* Qwen 2 - Direct Retrieval  *(b)* Qwen 2.5 - Direct Retrieval  *(c)* Qwen 3 - Direct Retrieval

*(d)* Qwen 2 - Indirect Retrieval  *(e)* Qwen 2.5 - Indirect Retrieval  *(f)* Qwen 3 - Indirect Retrieval

*(g)* InternLM 3 - Direct Retrieval  *(h)* InternLM 3 - Indirect Retrieval

*Figure 7.* Detailed performance breakdown for all Qwen and InternLM model families on the Direct Retrieval and Indirect Retrieval tasks.

# B. Detailed Pre-trained Models Results

This appendix provides details on the evaluation setup for the pre-trained model experiments presented in Section 2.

**Models.**  We evaluated four model families, comparing their text-only (Instruct) and vision-language (VL-Instruct) variants:

- **Qwen 2**: `Qwen2-7B-Instruct` and `Qwen2-VL-7B-Instruct`

- **Qwen 2.5**: `Qwen2.5-7B-Instruct` and `Qwen2.5-VL-7B-Instruct`

- **Qwen 3**: `Qwen3-8B` and `Qwen3-VL-8B-Instruct`

- **InternLM 3**: `InternLM3-8B-Instruct` and `InternVL3-9B`

**Inference Setup.**  All models were loaded using the HuggingFace Transformers library at full precision on a single NVIDIA RTX 3090 GPU. We used each model's default inference settings without modifying temperature or sampling parameters. Prompts were formatted using the model's chat template with the default system prompt ("You are a helpful assistant.").

**Evaluation Protocol.**    For each query, we checked whether the model's generated output contained the correct city name. We report mean accuracy and standard error across all query positions for each context length.

## C. Task Details

This appendix provides implementation details for the Indirect Retrieval task introduced in Section 3.

**Vocabulary.**    We define three disjoint vocabularies for the task:

- **Colors:** The full vocabulary consists of 216 colors from the web-safe color palette, formed by a $6 \times 6 \times 6$ RGB grid using hexadecimal values `00`, `33`, `66`, `99`, `cc`, and `ff` for each channel. During image-modality training, we restrict to the first 8 colors. In the text modality, colors are represented as tokens `color0001` through `color0216`.

- **Shapes:** We use 13 distinct shapes: rectangle, circle, hexagon, triangle, pentagon, rhombus, octagon, star, heart, semicircle, cross, arrow, and annulus. In the text modality, shapes are represented as tokens `shape0001` through `shape0013`.

- **Items:** Target labels are represented as tokens `item0001`, `item0002`, etc. Training uses up to 32 distinct items.

**Prompt Structure.**    Both modalities share a common prompt structure with the following special tokens:

- `[CONTEXTEND]` separates the context (attribute-entity pairs) from the association list.

- `[SEP]` separates individual entity-item associations.

- `[QUESTION]` marks the beginning of the query.

- `[ANSWER]` precedes the target output position.

An example text-modality prompt with 3 objects is:

```
color0113 shape0055 color0119 shape0041 color0153 shape0091
[CONTEXTEND] shape0041 item0029 [SEP] shape0055 item0025
[SEP] shape0091 item0020 [QUESTION] color0119 [ANSWER]
```

Here, the query asks for the item associated with `color0119`. Since `color0119` is paired with `shape0041` in the context, and `shape0041` maps to `item0029`, the correct answer is `item0029`.

In the image modality, the context is replaced by a sequence of image tokens extracted from a frozen encoder (e.g., 196 tokens per image for ViT-based encoders), while the associations and query remain in text.

**Image Rendering.**    For the image modality, each colored shape is rendered as a $224 \times 224$ pixel image. Shapes are drawn as filled SVG graphics on a white background. Figure 8 shows example renderings of colored shapes used as visual context.

**Feature Grounding Task.**    In addition to the main Indirect Retrieval task, we use an auxiliary *Feature Grounding* task to stabilize training and disentangle reasoning failures from perceptual failures. This simpler task requires the model to directly map an attribute to its corresponding entity without the intermediate item-retrieval step. Given a context of attribute-entity pairs and a query attribute, the model must output the associated entity.

An example text-modality prompt is:

```
color0022 shape0192 color0054 shape0209 [CONTEXTEND]
[QUESTION] color0022 [ANSWER]
```

The correct answer is `shape0192`. In the image modality, the context is similarly replaced by image tokens while the query remains in text.

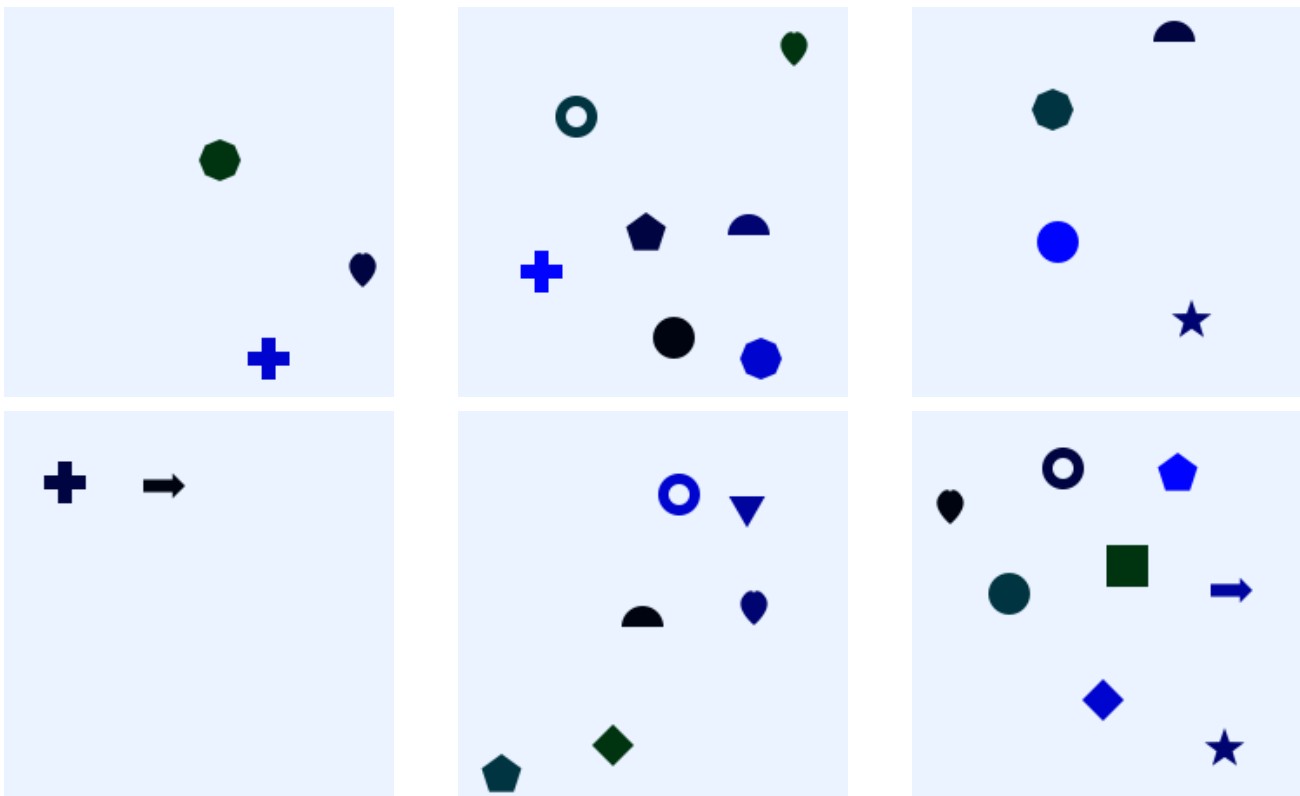

*Figure 8.* Example rendered contexts for the image modality, showing colored shapes drawn as filled SVG graphics on a white background.

**Symmetric Variations.** We consider symmetric variations of both tasks where the roles of entity and attribute are interchangeable. Specifically, we include both configurations: querying by color to retrieve a shape, and querying by shape to retrieve a color. For the Indirect Retrieval task, this extends to assigning items to shapes (retrieved via color) versus assigning items to colors (retrieved via shape). This symmetric formulation is used during training to increase data diversity. All evaluation results reported in the main text use the color-to-shape query direction.

## D. Detailed Train Procedure

In the previous section, we demonstrated that Vision-Language Models consistently outperform their text-only counterparts on retrieval tasks, particularly as context length increases. To understand the underlying mechanisms driving this phenomenon, we now replicate these findings in a controlled experimental setting. By training small transformers from scratch on our synthetic indirect retrieval task, we gain the ability to apply mechanistic interpretability tools and isolate the factors responsible for improved generalization.

### D.1. Model Architecture

We use a 12-layer decoder-only Transformer with the architecture detailed in Table 2. The model employs Rotary Positional Embeddings (RoPE) (Su et al., 2024) applied to the query and key projections, using a fixed base frequency of $\theta = 10000$ with interleaved rotation on even and odd dimension pairs. The vocabulary consists of approximately 1000 tokens, covering all color and shape combinations used in our experiments.

### D.2. Training Configuration

All models are trained using the AdamW optimizer with a learning rate of $1 \times 10^{-4}$, no weight decay, and gradient clipping with a maximum norm of 1.0. We use a batch size of 32 and train with bf16 mixed precision. The learning rate schedule consists of a 2000-step linear warmup followed by a constant learning rate. Training continues until validation performance plateaus, using early stopping. Experiments were conducted on NVIDIA A6000 and A40 GPUs.

*Table 2.* Model architecture hyperparameters.

| Hyperparameter | Value |
|---|---|
| Number of layers | 12 |
| Hidden dimension ($d_{\text{model}}$) | 128 |
| FFN dimension ($d_{\text{ff}}$) | 512 |
| Attention heads | 4 |
| Positional encoding | RoPE ($\theta = 10000$) |
| Dropout | 0 |
| Vocabulary size | $\sim$1000 |

### D.3. Text-Only Baseline

We begin by establishing a baseline model trained on the 1-hop indirect retrieval task defined in Section 3. We train exclusively on the text modality, where the context $\mathbf{X}^{\text{text}}_{\text{context}}$ consists of sequential token pairs representing attribute-entity associations (e.g., "red triangle", "blue circle").

Training a Transformer to solve this task from scratch is highly unstable due to the sparsity of the supervision signal. Since the intermediate tokens in the prompt are randomized or independent of the context, the model can only be trained to predict the final answer token. This means there is no direct reward for uncovering the intermediate logical steps (grounding and retrieval), requiring the model to discover the entire reasoning chain at once without partial credit for partial logic. To address this, we employ a five-stage curriculum to incrementally increase task complexity:

1. **Fixed Layout:** Two object pairs, restricted vocabulary (8 colors, 8 shapes), and fixed prompt order (Context $\rightarrow$ Bindings).

2. **Vocabulary Expansion:** Vocabulary increases to 216 colors and 216 shapes, while structure remains fixed.

3. **Variable Length:** Random sampling of 2 to 8 object pairs per prompt.

4. **Randomized Order:** Shuffling of pairs within the context and binding sections to prevent positional memorization.

5. **Scale:** Context capacity increased to support up to 32 items.

We denote the resulting model as $\mathcal{M}_{\text{text-only}}$. To ensure reliability, we repeated this full training curriculum using 4 distinct random seeds. For the experiments, we report the average performance across 4 seeds and the standard error (SE).

As shown in Figure 3, $\mathcal{M}_{\text{text-only}}$ (blue curve) generalizes poorly to OOD sequence lengths. Accuracy drops sharply beyond the training maximum of 8 items, resulting in an average OOD accuracy of 37.2% (SE: 4.0%).

### D.4. Noise Training

After completing the text curriculum, we continue training $\mathcal{M}_{\text{text-only}}$ with noise augmentation to extend its positional range. We insert 100 unattendable noise tokens between each context pair, exposing the model to longer positional indices without adding semantic information. The augmented context is formulated as:

$$\mathbf{X}^{\text{noise}}_{\text{context}} = [a_1, e_1, \underbrace{[\text{noise}], \dots, [\text{noise}]}_{100}, a_2, e_2, \dots, a_N, e_N]$$

This produces $\mathcal{M}_{\text{noise-text}}$. As shown in Figure 3 (orange curve), noise augmentation substantially improves OOD generalization, increasing average accuracy from 37.2% to 57.5% (SE: 6.8%).

Critically, noise training serves as a prerequisite for the image training pipeline described below. By extending the model's positional range, noise training enables the model to skip the curriculum stages when training on images, where patch tokens naturally produce longer sequences.

## D.5. Feature Grounding Pre-training

Before proceeding to image training, we train $\mathcal{M}_{\text{noise-text}}$ on the Feature Grounding auxiliary task in text mode. This task trains the model to predict associations in both directions: given an attribute, predict its associated entity (e.g., given "red", predict "triangle"), and vice versa. This establishes the structural requirements needed for the indirect retrieval task before introducing visual inputs.

## D.6. Image Training

Following the Feature Grounding pre-training, we train on the indirect retrieval task in the vision modality. Images are rendered at $224 \times 224$ resolution using SVG-based rendering, where each entity is depicted with its corresponding attribute (e.g., colored shapes on a canvas).

**Image Encoders.**    We experiment with three frozen pre-trained image encoders:

- **ResNet-152** (He et al., 2016): A supervised CNN producing 2048-dimensional spatial feature maps from the final convolutional stage. We discard global average pooling and fully connected layers, flattening the feature maps to produce pseudo-patch embeddings.

- **ViT-B/16** (Wu et al., 2020): A supervised Transformer producing 768-dimensional patch embeddings. We use features from the final layer, discarding the [CLS] token.

- **DINOv3** (Siméoni et al., 2025): A self-supervised Transformer producing 768-dimensional patch embeddings. We discard both [CLS] and register tokens to preserve spatial correspondence.

**Visual Projector.**    To map visual representations into the language model's embedding space, we employ a 3-layer MLP projector with a hidden dimension multiplier of $4\times$ (i.e., if the encoder outputs $d_{\text{enc}}$-dimensional features, the hidden layer has dimension $4 \cdot d_{\text{enc}}$). We maintain a strict one-to-one mapping where each visual patch is projected to a single token, without pooling. The resulting features are normalized via LayerNorm both before and after projection, and scaled by $1/\sqrt{d_{\text{model}}}$ before concatenation into the input sequence.

**Training.**    We train on image prompts, jointly optimizing for both 1-hop retrieval and Feature Grounding. For training stability, this stage uses a restricted distribution (8 colors, 13 shapes, up to 8 items), yielding $\mathcal{M}_{\text{image}}$.

## D.7. Text Transfer and Vocabulary Expansion

After image training, we transfer the model back to the text domain in two stages:

**Mixed Modality Training.**    We train on a 20/80 split of image and text tasks, maintaining the restricted vocabulary (8 colors, 13 shapes). This produces $\mathcal{M}_{\text{image-text-limited}}$.

**Vocabulary Expansion.**    Since the image training uses only 13 shapes (limited by what can be rendered distinguishably), we cannot directly evaluate OOD generalization to 32 items. To enable OOD evaluation, we continue training $\mathcal{M}_{\text{image-text-limited}}$ on text-only data, expanding the distribution to match the full complexity of the baseline (216 colors, 216 shapes, up to 32 items). This produces our final model $\mathcal{M}_{\text{image-text}}$.

**Noise Augmentation (Optional).**    To further improve generalization, we can apply noise training to $\mathcal{M}_{\text{image-text}}$, producing $\mathcal{M}_{\text{noise-image-text}}$.

## D.8. Experimental Setup

We conducted experiments using 4 random seeds. For each seed, we trained separate versions of $\mathcal{M}_{\text{image}}$ using each of the three image encoders (ResNet-152, DINOv3, ViT), resulting in a total of 12 experimental runs (4 seeds $\times$ 3 encoders). We report aggregated performance across 11 valid runs, excluding a single divergent run (Seed 2 with ViT encoder).

As shown in Figure 3, $\mathcal{M}_{\text{image-text}}$ (green curve) exhibits significantly improved performance over the text-only baseline, achieving an average accuracy of 69.5% (SE: 3.3%). This substantial improvement demonstrates that visual training enhances text-based generalization, replicating the phenomenon observed in large-scale VLMs.

### D.9. Analysis: Context Length vs. Visual Training

The superior performance of $\mathcal{M}_{\text{image-text}}$ raises a natural question: *Why does visual training improve text-based generalization?* One hypothesis is that the improvement stems from exposure to longer sequences during visual training. Since image encoders produce sequences of patch tokens (e.g., 196 tokens for a $14 \times 14$ patch grid), the model encounters a broader range of positional indices compared to text-only training.

To test whether context length exposure alone explains the performance gap, we compare models with and without noise augmentation (described in Section D.4). The noise-augmented text model $\mathcal{M}_{\text{noise-text}}$ achieves 57.5% average OOD accuracy, while the image-text model $\mathcal{M}_{\text{image-text}}$ achieves 69.5%. Applying noise augmentation to the image-text model yields $\mathcal{M}_{\text{noise-image-text}}$ at 83.6%.

Critically, even with noise augmentation, $\mathcal{M}_{\text{noise-text}}$ (57.5%) remains substantially worse than $\mathcal{M}_{\text{image-text}}$ without noise (69.5%). This demonstrates that while exposure to longer positional ranges helps, it is insufficient to fully explain the generalization advantages conferred by visual training. The visual modality must provide qualitatively different inductive biases beyond simple context length expansion.

### D.10. Key Findings

The controlled experiments presented in this section reveal several critical insights:

- Text-only models trained on short contexts exhibit poor OOD generalization (37.2% average accuracy on longer sequences).

- Visual training substantially improves generalization (69.5%), replicating the VLM advantage observed in large-scale models.

- Noise augmentation, which exposes models to longer positional ranges, provides substantial improvement to text-only models (57.5%), but remains insufficient to match multimodal performance.

- Even after noise augmentation, the vision-trained model maintains superior performance, suggesting that visual training induces qualitatively different computational mechanisms beyond positional range expansion.

- Combining visual training with noise augmentation yields the strongest performance (83.6%), suggesting complementary regularization effects.

These findings demonstrate that the benefits of multimodal training extend beyond simple exposure to longer sequences. The critical question remains: *What underlying computational mechanism does visual training induce that enables superior generalization?* In the following section, we employ mechanistic interpretability techniques to reveal that the answer lies in a fundamental shift in how models bind and retrieve information—transitioning from brittle positional strategies to robust identity-based binding mechanisms.

## E. Detailed Accuracy Sweeps (Scratch Models)

Figure 9 shows individual generalization curves for each scratch model variant. Models trained with meaningful visual input ($\mathcal{M}_{\text{image-text}}$) generalize beyond the training distribution, while text-only and noise-augmented variants show sharp accuracy drops at longer sequence lengths.

## F. Detailed Interchange Results

We performed interchange interventions across 4 random seeds for the Text-Only model and 11 runs (across 3 encoders) for the Image-Text model. Figure 10 illustrates the consistency of the mechanism type despite variations in circuit depth.

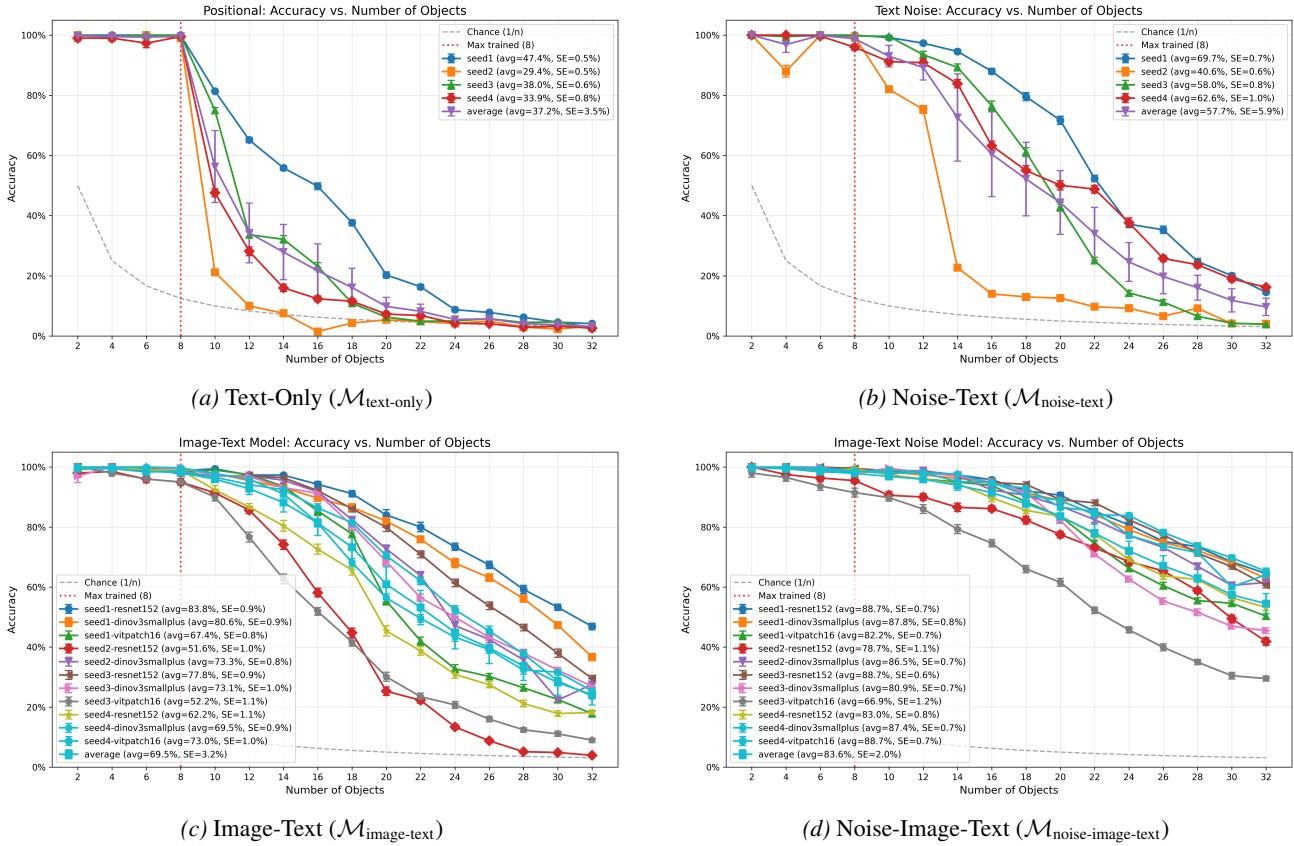

*Figure 9.* Individual generalization plots for the four scratch model variants. The vertical dashed line indicates the maximum sequence length seen during training (8 items).

It is important to note that the model can be marked as "Reflexive" if the answer is already stored in its activations from a previous step. In this context, the Reflexive signal represents a readout of stored information rather than the retrieval process. Therefore, the critical indicator of the binding mechanism is the *first* spike in causal effect.

### F.1. Global Binding Score

In subsection 5.6 in the main text, we report a **window-averaged binding score** that averages the symbolic-to-positional ratio across a designated binding window for each model, rather than relying on a single peak layer. This provides a more stable and representative measure of the dominant binding strategy.

Figure 5 shows the per-layer symbolic-to-positional ratio for all evaluated models; shaded regions indicate the exact binding window used for each model. Binding windows were selected by identifying the contiguous region around the peak intervention effect where the symbolic-to-positional ratio remained elevated above the baseline trend for each model. Table 3 lists the exact layer ranges. While the VLM variants consistently show higher ratios than their LLM counterparts, the exact layer of peak effect varies across models. To obtain a robust metric, we compute the average ratio within the designated window for each model. Figure 11 visualizes these window-averaged scores.

## G. Positional Invariance Experiment Details

To directly test the causal role of translation invariance in driving the binding mechanism shift, we trained models on a modified visual task where object positions were fixed across all images. In the standard visual task, object positions are sampled uniformly at random across the image canvas. In the fixed-position variant, each object type (e.g., red circle, blue triangle) always appears at the same spatial coordinates, eliminating translation invariance at the task level while keeping all other aspects of the training pipeline identical.

| MODEL | BINDING WINDOW |
|---|---|
| QWEN 2 | L19–L26 |
| QWEN 2.5 | L19–L26 |
| QWEN 3 | L23–L32 |
| INTERNLM 3 | L31–L42 |

*Table 3.* **Binding windows used for the window-averaged binding score.** Windows were chosen by inspecting the peak intervention effects.

We trained twelve independent models with different random seeds. After training, we evaluated their binding mechanisms using the same interchange intervention protocol described in Section 5 and Appendix F. Table 4 summarizes the results.

| SEED | ENCODER | DOMINANT MECHANISM |
|---|---|---|
| 1 | RESNET-152 | POSITIONAL |
| 1 | VIT-B/16 | SYMBOLIC |
| 1 | DINOV3 | SYMBOLIC |
| 2 | RESNET-152 | SYMBOLIC |
| 2 | VIT-B/16 | POSITIONAL |
| 2 | DINOV3 | POSITIONAL |
| 3 | RESNET-152 | POSITIONAL |
| 3 | VIT-B/16 | SYMBOLIC |
| 3 | DINOV3 | SYMBOLIC |
| 4 | RESNET-152 | SYMBOLIC |
| 4 | VIT-B/16 | POSITIONAL |
| 4 | DINOV3 | SYMBOLIC |

*Table 4.* **Binding mechanisms with fixed object positions.** Across twelve random seeds, five models retained positional binding while seven transitioned to symbolic binding, indicating that translation invariance contributes to but does not fully explain the mechanism shift.

The mixed results suggest that pretrained image encoders retain translation-invariant features from their architecture and pretraining, even when spatial variability is reduced at the task level. Consequently, other properties of visual training likely also contribute to the shift toward symbolic binding.

## H. Detailed Circuit Analysis

This section provides detailed analysis of how each binding mechanism operates at the circuit level. We use attention knockouts to identify critical token-to-token connections and linear probes to trace information flow through the network. Together, these reveal the computational structure underlying each mechanism.

### H.1. Methodology

**Attention Knockouts.** For each source-target token pair, we mask all attention from the source to the target and measure the resulting accuracy drop. High knockout effect indicates that the information flow between those tokens is essential for the task.

**Linear Probes.** We train lightweight classifiers on residual stream activations to decode position indices, color identity, shape identity, and item identity at each layer and token position. By tracking when and where each type of information becomes decodable, we trace semantic content through the network.

### H.2. The Positional Circuit

The text-only model implements binding through position indices rather than semantic content. The knockout and probe results reveal how this works in detail (Figures 12a and 13a).

**Step-by-Step Information Flow.** The circuit operates through two independent streams that converge at the answer:

1. **Query Stream:** The question token (the color in the query) attends to the matching color in the context section. It

copies the *position index* of that context color into its own activations. The [ANSWER] token then copies this position from the question token next to it.

2. **Association Stream:** Independently, the shape token in the association section attends to the matching shape in the context section and copies its position index. The item token in the association section then copies this position from the shape token next to it.

3. **Retrieval:** The [ANSWER] token looks for the item in the association section that holds the same position index it is holding. When the positions match, retrieval succeeds.

**Evidence from Knockouts.** The knockout matrix confirms these critical pathways: Question → Context Color (position lookup), Answer → Question (position copy), Association Shape → Context Shape (position lookup), and Association Item → Association Shape (position copy). Notably, there is no critical attention between color and shape tokens—the circuit never routes semantic content between positions.

**Evidence from Probes.** The probes show position information accumulating at each step: first at the question and association shape tokens (after the lookup), then at the answer and association item tokens (after the copy). Critically, attribute identity (color, shape) remains *undecodable* at entity positions throughout—the model never represents "red circle" as a bound unit. Only position indices are transferred.

This explains why the positional mechanism fails on longer sequences: it relies on position counting, and unfamiliar position indices break the heuristics.

### H.3. Symbolic Circuit A (Color-Key)

Image-trained models using the Color-Key variant implement explicit attribute binding (Figures 12b and 13b). Unlike the positional circuit, semantic content—not just position indices—is transferred between tokens.

**Step-by-Step Information Flow.** The circuit operates through two streams that converge at the answer:

1. **Binding in Context:** The context shape token accumulates the *color identity* from the adjacent context color token. This creates an explicit attribute-entity bundle: the shape now "knows" its color.

2. **Association Stream:** The shape in the association section looks back at the matching shape in the context section and copies the color into its activations. The item in the association section then copies the color from the adjacent shape.

3. **Query Stream:** Independently, the [ANSWER] token copies color from the adjacent query (color token).

4. **Retrieval:** The [ANSWER] token looks for the item in the association section that has the same color in its activations. When the colors match, retrieval succeeds.

**Evidence from Knockouts.** The knockout matrix confirms these attribute-routing pathways: Context Shape → Context Color (binding), Association Shape → Context Shape (color retrieval), Association Item → Association Shape (color propagation), and Answer → Query (color copy). These pathways are absent in the positional circuit.

**Evidence from Probes.** The probes reveal the **binding signature**: color identity first becomes decodable at the context shape token (binding), then at the association shape and item tokens (retrieval and propagation). This confirms that semantic content is being actively moved between tokens.

### H.4. Symbolic Circuit B (Shape-Key)

The Shape-Key variant uses a different intermediate representation but achieves the same explicit binding (Figures 12c and 13c).

**Step-by-Step Information Flow.** The circuit again operates through two streams:

1. **Binding in Context:** As in Circuit A, the context shape token accumulates color identity from the adjacent context color.

2. **Association Stream:** The item in the association section copies the adjacent *shape identity* into its activations.

3. **Query Stream:** Independently, the query (color token) looks back at the context and finds the shape marked with the same color. It copies the *shape identity* into its activations. The `[ANSWER]` token then copies this shape from the adjacent query.

4. **Retrieval:** The `[ANSWER]` token looks for the item in the association section that has the same shape in its activations. When the shapes match, retrieval succeeds.

**Evidence from Knockouts.** The critical pathways differ from Circuit A: Query → Context Shape is now critical (the query retrieves shape identity), and Association Item → Association Shape (shape propagation).

**Evidence from Probes.** The binding signature appears with shape identity rather than color. Shape information becomes decodable at the query token after attending to the context shape, then at the `[ANSWER]` token. Despite the different intermediate key, the fundamental property is preserved: semantic content transfers between tokens, enabling length-agnostic retrieval.

### H.5. Consistency Across Seeds and Encoders

The patterns are consistent across experimental runs (Figures 14 and 15). All text models exhibit the positional pattern; all image-text models exhibit the symbolic pattern. The specific variant (A vs. B) varies by seed and encoder, but explicit attribute transfer is universal across image-trained runs. This confirms that symbolic binding is a robust emergent property of vision-language training.

### H.6. Extension to Large-Scale Models

We extended our analysis to the Qwen model family (Qwen 2, 2.5, and 3). Full circuit analysis via attention knockouts is computationally prohibitive at this scale, so we use linear probes as a proxy: if a model implements symbolic binding, attribute information should become decodable at entity positions.

The VLM variants exhibit the binding signature. Color decodability at entity tokens is consistently higher in VLMs than in their text-only counterparts across the layer ranges where binding occurs. Moreover, the *ordering* of when probes rise coincides with the proposed order of Symbolic Circuit A: color information first becomes decodable at the context shape token, then at the association shape token, reflecting binding in context before propagation through associations. Text-only baselines show low attribute decodability at these positions throughout.

See Appendix K for detailed results. The consistency across model scales suggests symbolic binding is a general property of vision-language training.

## I. Linear Probe Analysis

In this section, we provide the aggregate linear probe results across all runs.

**Probe Training Details.** We use a standard linear probing setup to measure representational content without introducing additional modeling capacity. Architecture: a single linear layer, `nn.Linear`$(d_{\text{model}}, \text{num\_classes})$. Training: frozen activations with an 80/20 train–validation split, trained using cross-entropy loss and Adam (learning rate = 0.01) for 100 epochs. Rationale: training independent probes per layer allows us to identify when semantic attributes (e.g., color identity) become linearly separable in the residual stream.

See Appendix H for the aggregate linear probe and knockout visualizations.

## J. Attention Knockout Analysis

We present the aggregate attention knockout results. These heatmaps show the average impact on task performance when individual attention heads are masked, averaged across all seeds for the respective model types. See Appendix H for the visualizations and detailed circuit-level interpretation.

## K. Qwen Probe Results

We present detailed linear probe results for the Qwen model family. Figure 16 shows color decodability at the context shape token, where binding first occurs. Figure 17 shows color decodability at the association shape token, reflecting propagation of bound attributes. Across all three model generations, VLMs consistently show higher color decodability at entity positions than their text-only counterparts.

## L. Related Work (Extended Version)

**Multi-modal training effects on language models.**   Multi-modal training has been studied by comparing multi-modal models against their original unimodal backbone language models in order to assess how augmenting them with visual modalities alters their behavior and performance. Several works provide theoretical grounding suggesting that multi-modal training promote robustness and generalization beyond the training distribution, arguing that it discourages reliance on spurious, modality-specific correlations that are over-represented in unimodal training data (Xue et al., 2024) and instead promote more faithful estimation of the underlying latent semantic space (Huang et al., 2021). Empirically, vision–language models (VLMs) have been shown to retain, and in some cases even improve, strong performance on purely textual tasks. For instance, Dai et al. (2024b) demonstrate that, under appropriate training regimes, a VLM can preserve or enhance the text-only capabilities of its language backbone across benchmarks in language understanding, mathematics, coding, and reasoning, while Ratzlaff et al. (2025) reporting similar findings on commonsense reasoning. However, these effects are highly model- and backbone-dependent: Ratzlaff et al. (2025) show that for LLaVA-style models, multi-modal training can either improve or degrade textual performance depending on the underlying LLM and, in some cases, introduce degradation in mathematical reasoning. Complementarily, studies have also examined the impact in the opposite direction, from language to vision. Cooper et al. (2025) find that while pure VLMs outperform hybrid VLM–LLM systems on perceptual tasks such as object and scene recognition, the inclusion of an LLM yields gains on tasks requiring higher-level reasoning or external knowledge.

**Mechanistic interpretability**   Mechanistic interpretability provides a principled framework for analyzing how neural networks implement computations internally, enabling causal rather than purely correlational explanations of model behavior. In this work, we adopt a suite of complementary interpretability tools to characterize and compare the mechanisms underlying the models we study, before and after image training. First, we employ interchange interventions (Meng et al., 2022; Geiger et al., 2021; Finlayson et al., 2021; Vig et al., 2020; Geiger et al., 2020), which causally intervene on hidden states by swapping representations between paired examples, allowing us to identify which internal activations are responsible for specific outputs. This approach is closely related to recent studies on binding and compositional representations in Transformers (Feng & Steinhardt, 2024; Saravanan et al., 2025; Gur-Arieh et al., 2026; Wu et al., 2025), and enables fine-grained analysis of how information is localized and propagated through the network. Second, we use attention knockout techniques (Geva et al., 2023; Gur-Arieh et al., 2026), which selectively zero out attention connections between targeted token pairs, to evaluate how disrupting specific information pathways affects task performance. Finally, we rely on linear probing methods (Alain & Bengio, 2017; Ravichander et al., 2021; Belinkov, 2022), training lightweight classifiers on hidden representations to assess whether particular concepts or variables are linearly decodable, thereby providing insight into what information is encoded at different layers of the model. Together, these tools allow us to causally and representationally dissect the internal computations of Transformers and to connect observed generalization behavior with concrete underlying mechanisms.

**Binding**   Binding refers to a model's ability to correctly associate entities with their corresponding attributes (Treisman, 1996; Feng & Steinhardt, 2024). Recent work has investigated the internal mechanisms by which Transformers implement binding, typically through controlled retrieval tasks that require recovering an attribute given an entity in the query (Feng & Steinhardt, 2024; Saravanan et al., 2025; Wu et al., 2025; Gur-Arieh et al., 2026; Prakash et al., 2026). These studies intervene on model activations to characterize the information used to form bindings and how these mechanisms evolve during learning, discovering different behaviors that this mechanism shows through training (Wu et al., 2025). Using the taxonomy introduced by Gur-Arieh et al. (2026), prior work, distinct binding strategies in language models have been identified, including positional mechanisms that rely on token order or relative position (Dai et al., 2024a; Prakash et al., 2024; 2026), symbolic (termed lexical in the original work) mechanisms that exploit content-based cues, and reflexive mechanisms that use a self-referential association between tokens (Gur-Arieh et al., 2026). Complementary analyses based on attention patterns further support this view, with Urrutia et al. (2026) providing evidence for both positional and symbolic

binding heads in language models trained on retrieval tasks similar to ours. Moreover, detailed studies of attention head behavior reveal the emergence of specialized heads that systematically route information across token positions to facilitate binding operations (Wu et al., 2025).

**Length generalization**  A substantial body of work has demonstrated that neural networks often struggle with length generalization, i.e., extrapolating to input lengths longer than those observed during training, and has introduced benchmarks to systematically evaluate this capability (Lake & Baroni, 2018; Bastings et al., 2018; Ruis et al., 2020; Zhang et al., 2022; Saparov et al., 2023; Zhou et al., 2024a). In the context of Transformers, these limitations have been observed across a variety of settings, including arithmetic tasks such as summing longer numbers (Zhou et al., 2024b), varying the length of chains of reasoning (Zhang et al., 2022), deductive reasoning (Saparov et al., 2023), and general algorithmic problems (Zhou et al., 2024a). Nevertheless, several studies indicate that Transformers can exhibit weak but non-negligible extrapolation under specific conditions. For instance, pre-training has been shown to improve robustness to length extrapolation (Zhang et al., 2022), while architectural choices such as relative positional encodings (Csordás et al., 2021), NoPE (Shen et al., 2023), and the integration of FIRE and randomized positional encodings (Zhou et al., 2024b) can mitigate positional overfitting. Complementarily, carefully designed input representations that better align with the underlying task structure have been shown to facilitate length generalization (Shen et al., 2023; Zhou et al., 2024b). Finally, even minimal exposure to longer sequences during training, through the inclusion of a small number of long examples, can substantially improve extrapolation performance (Jelassi et al., 2023).

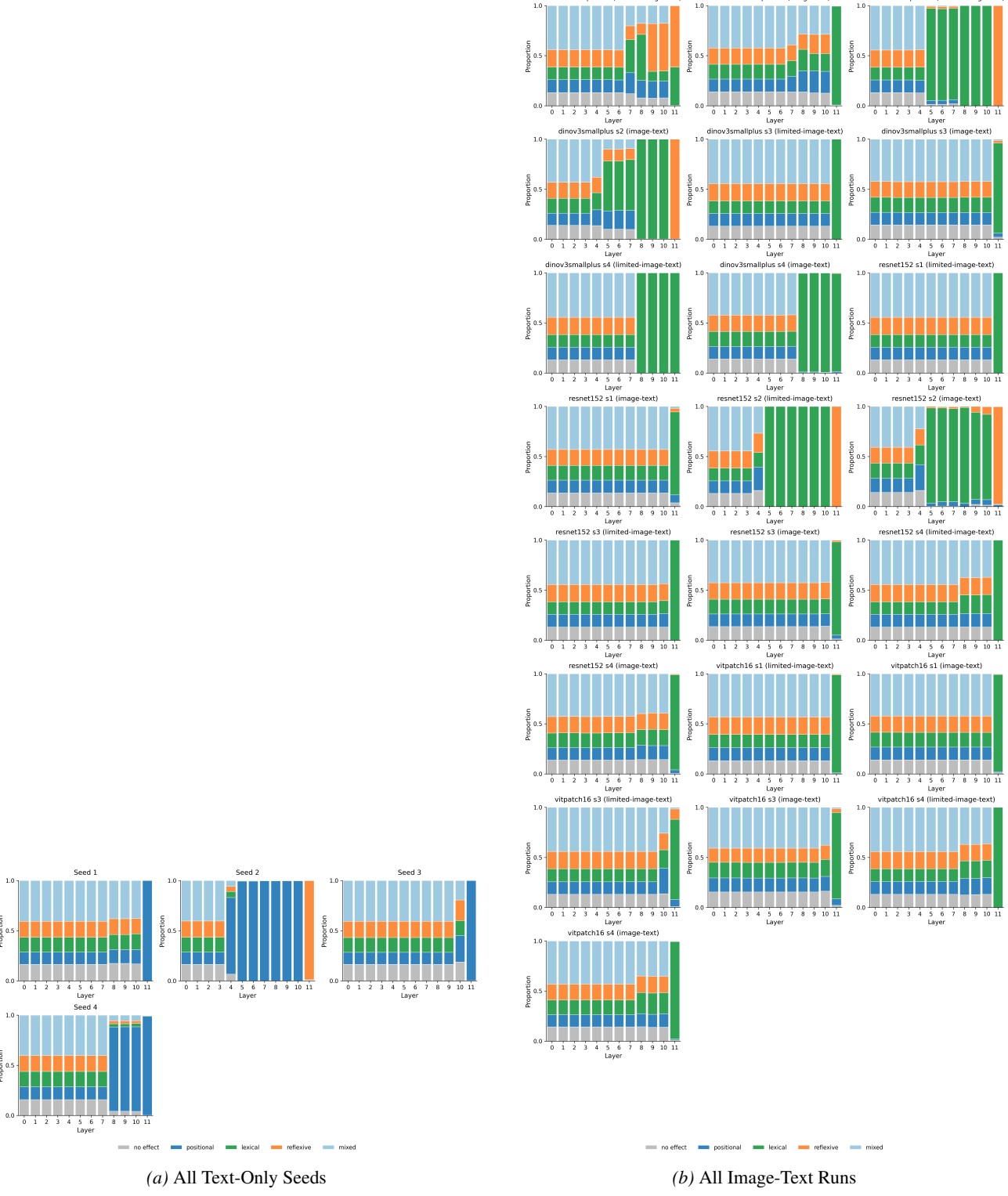

*(a)* All Text-Only Seeds

*(b)* All Image-Text Runs

*Figure 10.* **Aggregate Interchange Results.** (a) Across all 4 random seeds, the text-only models consistently converge to a Positional Context mechanism, though the specific layer where this mechanism activates varies (e.g., Layer 5 vs Layer 11). (b) Conversely, models exposed to the visual curriculum consistently develop Symbolic binding mechanisms, regardless of the specific image encoder used (ResNet vs ViT vs DINO).

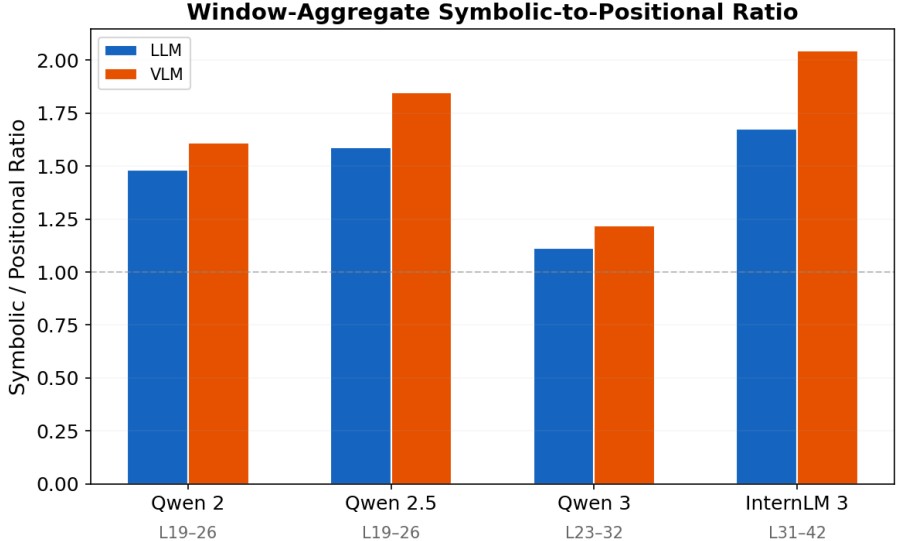

Figure 11. **Window-averaged binding scores.** For each model, we average the symbolic-to-positional ratio across the binding window indicated in Figure 5. This aggregated metric confirms a consistent shift toward symbolic binding in VLM variants across all tested families.

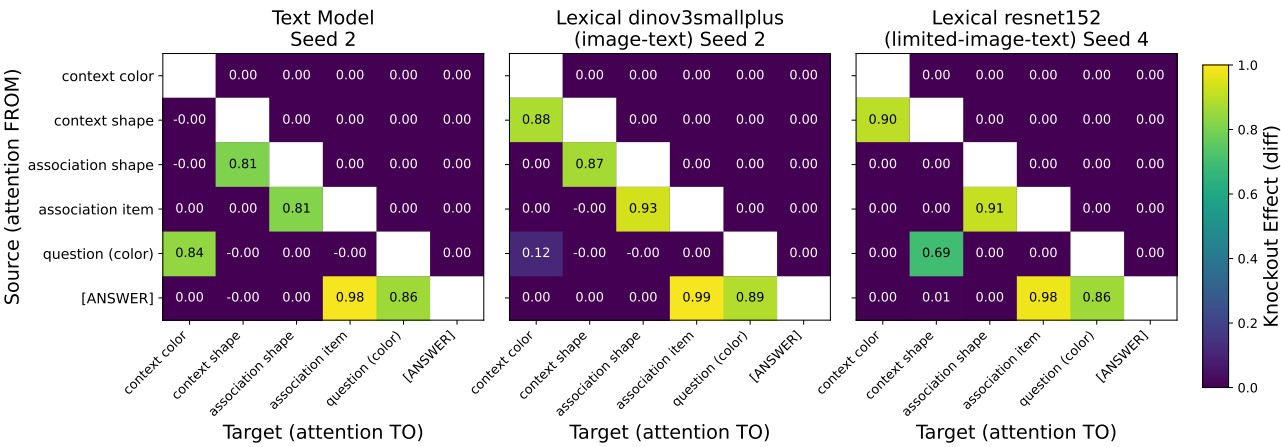

Figure 12. **Attention Knockout Analysis.** Heatmaps showing the drop in performance when attention between specific token pairs is ablated. (a) The Positional mechanism relies on position-matching pathways. (b, c) The Symbolic mechanisms show attribute-routing pathways.

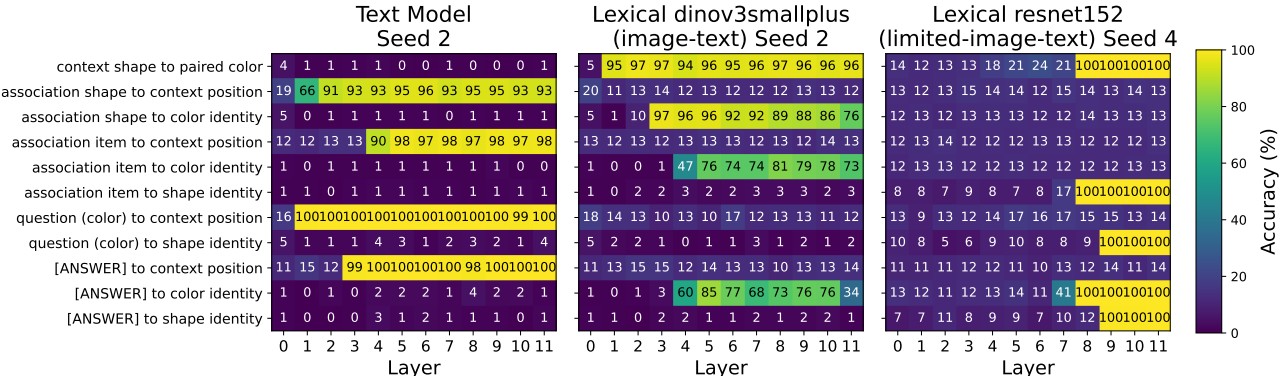

Figure 13. **Linear Probe Analysis.** (a) The Positional model accumulates position but not attribute information at entity tokens. (b, c) Symbolic mechanisms exhibit the "binding signature": attribute decodability surges at entity positions.

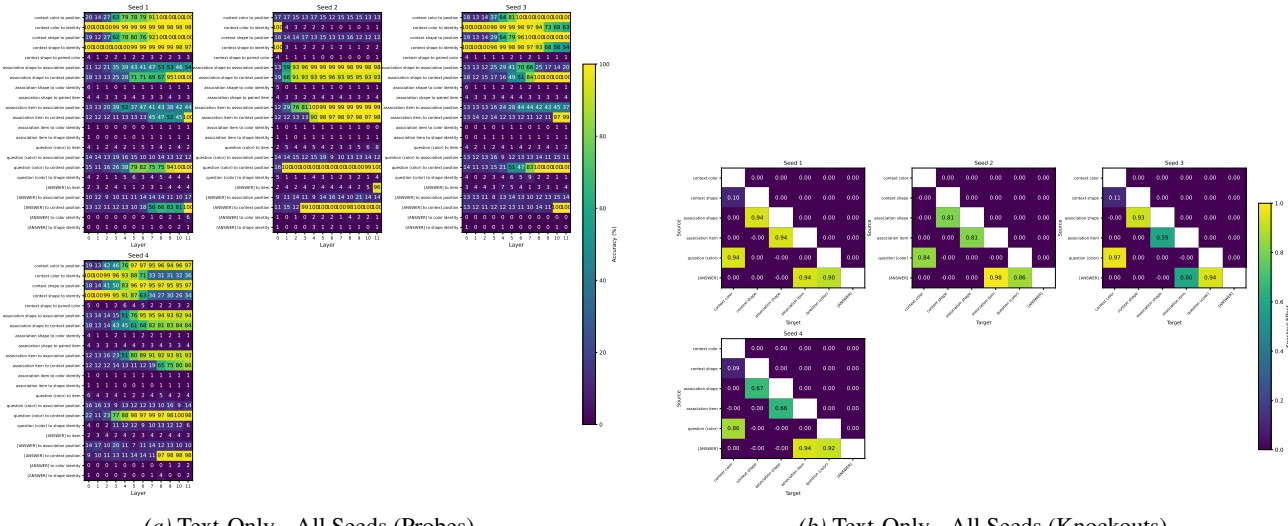

*(a)* Text-Only - All Seeds (Probes)        *(b)* Text-Only - All Seeds (Knockouts)

*Figure 14.* **Aggregate Results: Text Models.** All four seeds show the positional pattern: high position decodability, low attribute decodability at entities, and sparse position-matching knockout matrices.

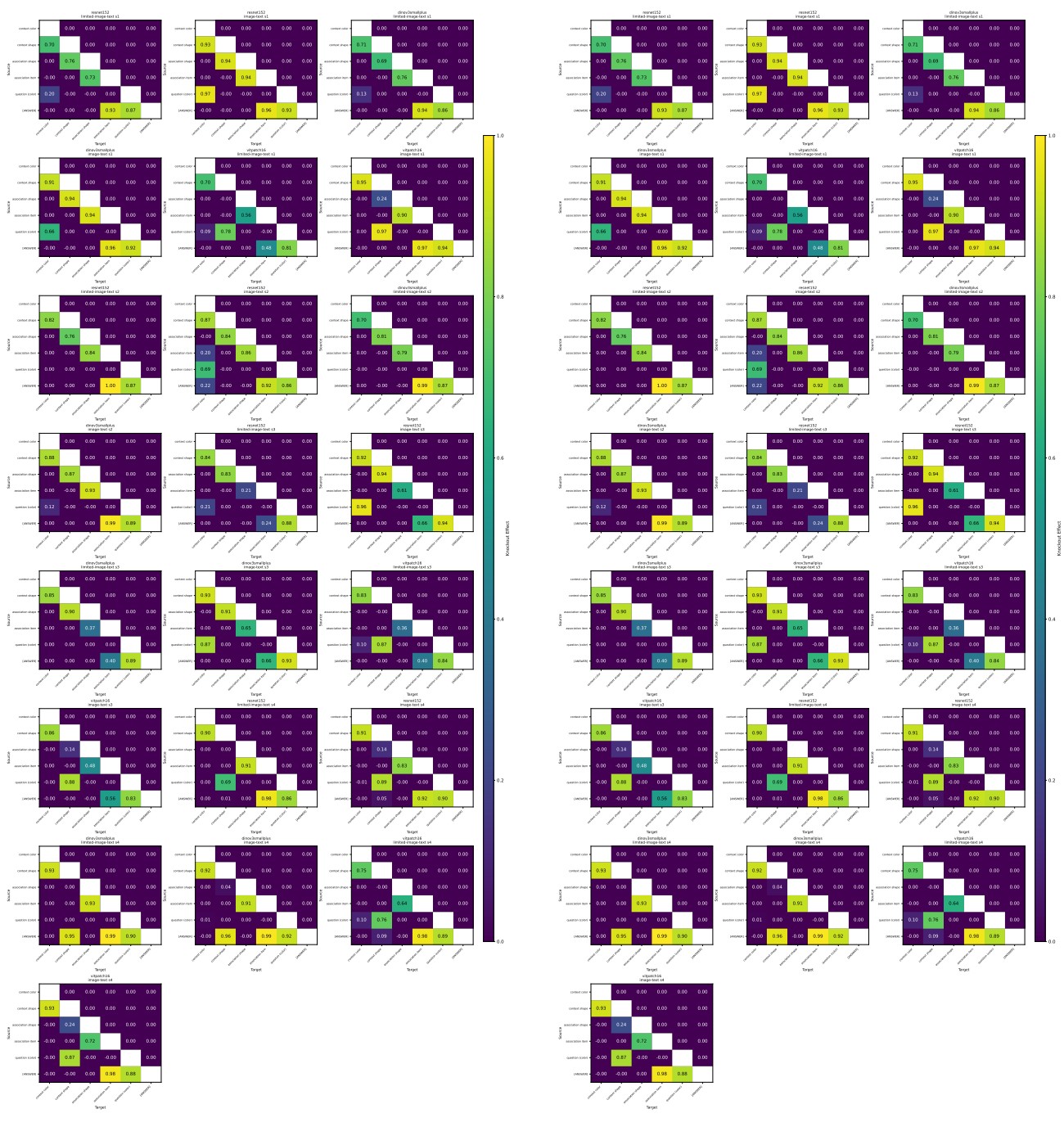

*(a)* Image-Text - All Runs (Probes)  *(b)* Image-Text - All Runs (Knockouts)

*Figure 15.* **Aggregate Results: Image-Text Models.** All eleven valid runs (4 seeds × 3 encoders, excluding one divergent run) show the symbolic pattern: binding signature in probes and attribute-routing pathways in knockouts.

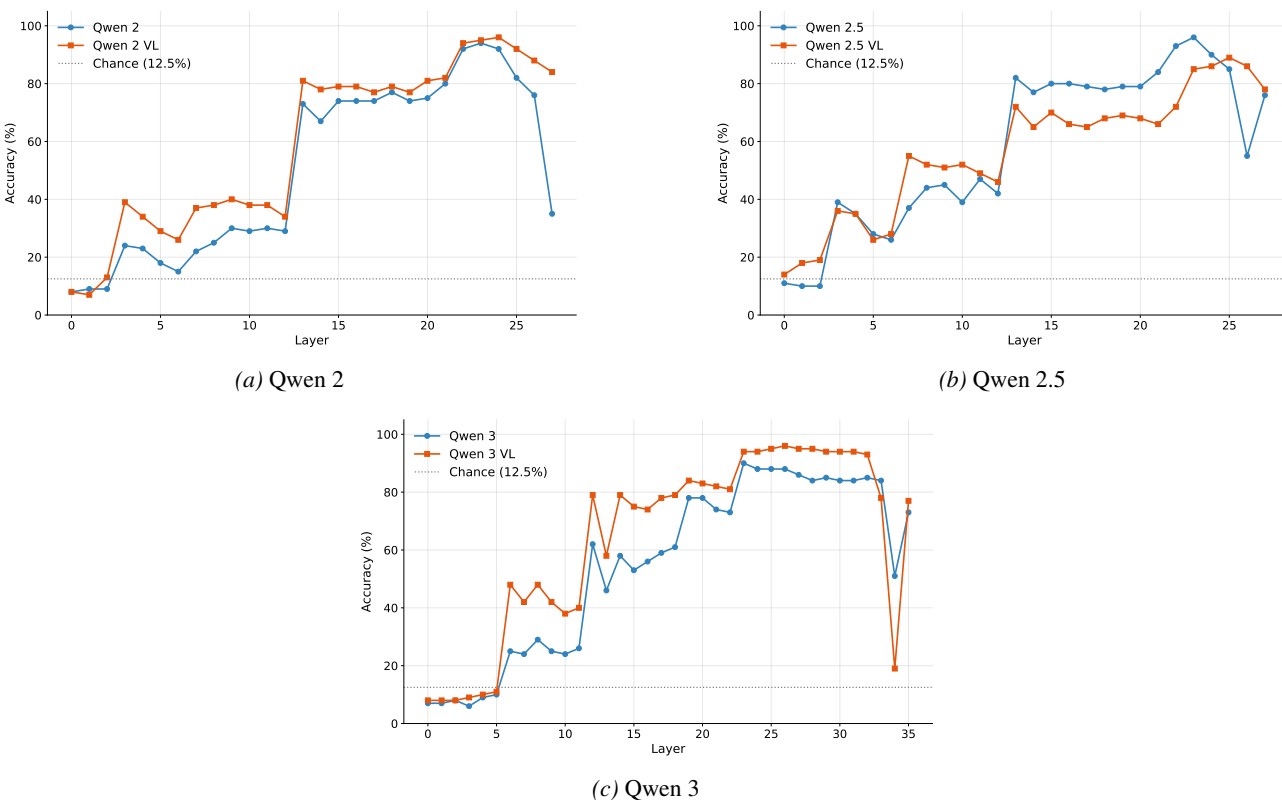

*(a)* Qwen 2

*(b)* Qwen 2.5

*(c)* Qwen 3

*Figure 16.* **Color at Context Shape.** Linear probes decoding color identity at the context shape token position. The VLM models show higher color decodability than their text-only counterparts, indicating binding in context.

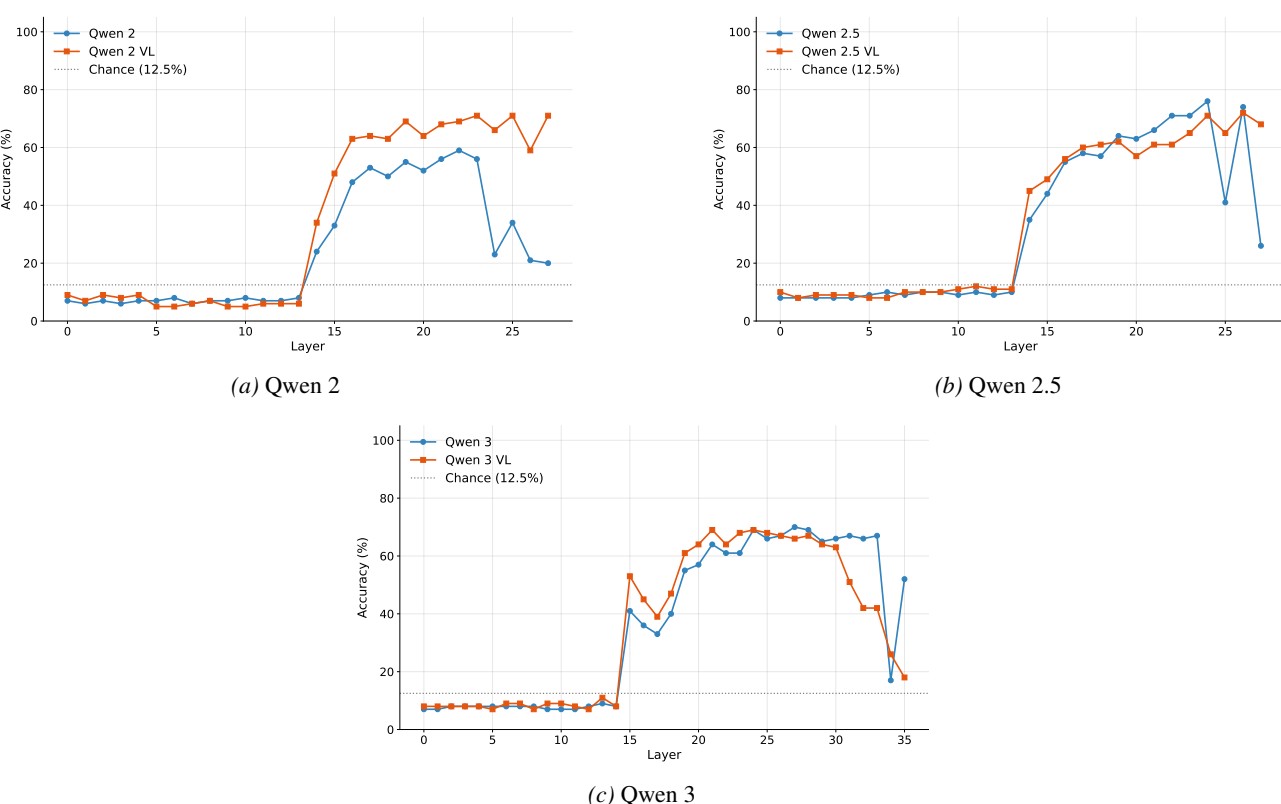

*(a)* Qwen 2

*(b)* Qwen 2.5

*(c)* Qwen 3

*Figure 17.* **Color at Association Shape.** Linear probes decoding color identity at the association shape token position. The VLM models show higher color decodability, consistent with color propagation from context to association.

