# OpenReview forum: "Seeing to Generalize: How Visual Data Corrects Binding Shortcuts"
_ICML.cc/2026/Conference — ICML 2026 regular_

### Official Review · Reviewer_YuhG · 2026-02-25

**Soundness:** 3
**Presentation:** 4
**Significance:** 3
**Originality:** 3
**Overall Recommendation:** 6
**Confidence:** 5

**Summary:**

The authors observe that VLMs outperform their underlying pre-trained LLMs on text-only tasks, in particular for out-of-distribution (OOD) data. They thus investigate this question and identify that, while LLM use positional shortcut to help their reasoning, adding visual task allows the model to rely on more symbolic shortcuts, leading to a better OOD generalization. The authors start by illustrating the initial observation with a toy experiment. Then, this one is run "at scale" with three couple of large LLM+VLM (namely Qwen 2, 2.5 and 3) and confirm the obsevation. They further explore the binding mechanism of the models to explain the different behaviors and find LLM mainly rely on positioning while VLM act at a more symbolic level. Then, they use attention knockout (Geva et al, 2023) to establish circuit architecture, in the vein of Mechanistic Interpretability works. It allows to better understand how the binding mechanism previously identified are actually implemented into the LLM/VLM.

**Compliance With Llm Reviewing Policy:**

Affirmed.

**Final Justification:**

Following the rebuttal I intended to raise my score from 4 to 5. Considering the originality of the approach, and the fact that other reviewer did not noticed any flaw as well, I do not see any reason  to not raise my score up to 6.

Justification:
* the paper is well written (actually pleasant to read) and technically perfectly sound
* the idea is original *and* can be a source of inspiration for further works in Mechanistic Interpretability.
* the work relies on available material and should be easy to reproduce (thus enforcing possible future works building on it)
* it provides new insights and deepen understanding of large models

**Key Questions For Authors:**

Some question that are likely to influence my final rating are listed below. Some of them are minor points but I am interested by the answer. At the opposite **questions 3 and 5 are important** for my final opinion. If it confirms there is no problem, I will raise the recommandation.

0/ will the authors release some material (python scripts) to reproduce the experiments if the paper is accepted? Although the paper contain enough details to reproduce it, it would help further research on the topic (and probably increase the impact of the paper)

1/ would it make sense to report the OOD performance for the experiment of line 2 in Figure 1 ?

2/ how the the context lengh is changed in the experiment of Section 2 (variable $l$ in line 159-left) ? Why the figure 2 mention "number of relations" rather than "context length"?

3/ to which extent a Qwen LLM and it corresponding VLM are fairly comparable in terms of (i) capacity (number of parameters) (ii) training dataset (size, "quality", "diversity") used for their pre-training ?

4/ line 191: the context is said to be "a sequence of image". Is it not rather "a sequence of image *tokens*" ? If not, it should be better explained

5/ how the position of the colored object is fixed in the images? To which extent do it varies? cf. line 320-left.

6/ how the "lightweight classifier" (line 931 in appendix G) used to implement the linear probes are trained ?

**Limitations:**

The authors use the standard impact statement, that makes sense for the submitted manuscript.

Beyond mentioning possible future work in the last sentence, the limitations of the approach are not further discussed.

**Strengths And Weaknesses:**

**Soundness**

The main experiments are conducted with three types of Qwen models, that are open weight and can be easily downloaded from huggingface. Moreover, it insures that the models are "freezed" and that their performance will not vary in time (contrary to API-only models), which is a positive point for the reprocucibility. The choice of the Qwen model is well justified by the fact that each model exists both as a LLM and a VLM.

The protocol of experiments in Section 2 (text only) is quite simple and the main details are reported in the text and the appendix to reproduce the experiments. Beyond some details (see below 'minor and details") this experiment clearly shows "at scale" the initial observation of the toy experiment (VLM outperform their corresponding LLM on text-only tasks) and is convincing. The only reserve I have concerns the relative size/capacity of both models. While unlikely, if the VLM is significantly larger (in terms of number of parameters) than the corresponding LLM, it could be a reason to have better performance. It could also be linked to training datasets (to pre-train the VLM/LLM) that are different and maybe larger/richer for the pre-trained VLM.

The experiment of section 3+4 deals with the use of additive visual information. The choices made to design the experiment make sense. The difference between in-domain samples (8 colors, 13 shapes,
8 items) and OOD ones (216 colors, 216 shapes, 32 items) is quite impressive, and the results (Fig. 3) are thus convincing. One appreciates that all experiment are run with several random seeds and averaged, and that the standard deviation is reported in the Figure (it was also the case in section 2). The control experiment with noise tokens inserted in the sequence is also valuable and addresses a question that naturally comes to mind for the reader.

Section 5 investigate the actual nature of variable binding within the models .The experiment is based on the method recently proposed by Gur-Arieh et al. (2025). While rapidly explained, it remains superficial, even in Appendix F: if one wants to reproduce it, it would require to read the original paper. The results are nevertheless a major contribution of the paper, giving convincing hints on the reasons why VLM better generalize than LLM on text-only tasks. The argument relating to the fixed/variable position of objects (section 5.4) is interesting and partially supported by the experiment with noise token inserted (section 5.5). However, it mainly modify the text modality, it would have been interesting to conduct an experiment with objects at fixed position as well.

The last experiments (section 6) go further by showing more directly the binding process through circuits. The methodology mainly relies on attention knockout, that is relevant. The results provides a deeper understanding of the models considered.

**Presentation**

The paper is very well written, with a clear structure. I do not write it often but it worth to mention it here: it is a pleasure to read this manuscript. In particular, the reader appreciates the presentation of the motivation through the initial toy experiment. As a reader, I also appreciate that the related works are placed at the end of the article. It really makes sense in this context since it promotes the fluidity of reading.

The related works are well structured (section 7) and even much more detailed in appendix I. The topic are relevant but it misses an explicit positioning of the work, explaining the precise contribution of the proposed work with regards to these previous works.

**Significance**

This paper addresses important problem relating to state of the art models, massively used in machine learning. It is very pleasant to read, and makes you want to know more and contribute to better understand some fundamental aspect of LLM/VLM. Beyond the initial observation, the experiments and their results are interesting and inspiring: I have few doubt it can be a source of inspiration for further works in Mechanistic Interpretability.

**Originality**

Comparing directly a LLM and its corresponding VLM is original. While the idea may seem simple (once someone had it...) the authors showed it is fruitful.

The work partially relies on existing methods to conduct their analysis. However, it is relevant for each experiment and explored to show novel results. It provides new insights and deepen understanding of large models. If the paper is accepted, I think the "previous method used" will become even more popular for conducting this type of research.

Globally, the paper shows that one can conduct good and inspiring research with existing models and (partially) existing methods.

**Minor and details**
- it would be nice to report the exact settings of the initial toy experiment in an appendix: which transformer is used? Do the prompt are exactly those of Figure 1 / page 4?
- line 108 right: it makes sense the $n_i$ appears only once but for $c_i$ they could appear several times (two persons could live in the same city)
  - line 141: same thing for the food: two persons could like the same food. However, in that case, the task could not be the same since the answer to a question could contain several cities.
- on lines 158-162 (left) the main text mention "context length" but refer to Figure 2 where the X-axis is the "number of relation". Is it the same? A typo?
  - as well, it is not clear how the "context length" can vary. One guesses that the formulation of the context changes but the exact way it is done is not explained. It would thus be difficult to exactly reproduce the same result.
- I suggest to add a number to the "equations" (prompts) on page 4. It would be easier to refer to them as a reviewer. However, for a final version of the manuscript, it may be useless.
- section 4 report the results of the experiment in section 3. Hence, they could merged.
- line 2025-left: the "contexts contain up to 8 objects" is unclear. Does the training start with training sample with less than 8 objects?
- line 346-left: "As shown in Section 5.6" --> "As shown in Table 1"
- a reference to (Geva et al, 2023) should be added on line 367, after "attention knockout".
- a reference to appendix I in section 7 would be nice.

---

> ### Author Rebuttal · Authors · 2026-03-31
>
> We sincerely thank the reviewer for this thorough, thoughtful, and highly constructive review. We greatly appreciate the time and care taken to evaluate our work in depth, as well as the detailed feedback and insightful questions. We are especially grateful for the positive assessment of the paper’s clarity, motivation, experimental design, and significance, and we are encouraged that the reviewer found the results convincing and inspiring for future work in mechanistic interpretability.
>
> **0. Will the authors release the code if the paper is accepted?**
>
> Yes. We fully agree with the importance of reproducibility and are committed to releasing the complete codebase upon acceptance.
>
> **1. Would it make sense to report the OOD performance for the experiment of line 2 in Figure 1?**
>
> Yes, this makes sense. However, our focus was on understanding how visual training improves OOD generalization in the original text modality, and we therefore did not evaluate OOD performance in the image modality.
>
> In our controlled setup, we kept the image modality simple, using only 8 distinct shapes so that each appears at most once per image, avoiding ambiguity. This design choice prevents constructing image contexts longer than 8 objects, making direct OOD evaluation in the image modality infeasible.
>
> Evaluating OOD generalization for images would require expanding the shape vocabulary (e.g., to ~32 shapes), which is an interesting direction for future work.
>
> **2. How is the context length changed in Section 2?**
>
> In our synthetic tasks, the number of relations corresponds to the number of entity–attribute associations in the context (e.g., “Person A lives in City X”). We use this quantity because it directly reflects the reasoning complexity of the retrieval task. Since each relation consists of a nearly fixed number of tokens, the number of relations is highly correlated with context length. We will clarify this explicitly in the manuscript to avoid confusion.
>
> **3. To which extent are Qwen LLMs and their corresponding VLMs comparable in capacity and training data?**
>
> The Qwen-VL models are built directly on the same transformer backbones as their text-only counterparts, sharing identical architecture and parameter count (considering the text backbone used in our experiments). The VLMs are initialized from the pre-trained LLM weights and subsequently undergo multimodal instruction tuning. While the final tuning data differs to incorporate visual inputs, the underlying linguistic representations are identical.
>
> Importantly, this phenomenon is not specific to Qwen. During the rebuttal period, we reproduced the same qualitative results using the InternLM‑3 (LLM) and InternVL‑3 (VLM) family, further supporting the generality of our findings (see answer to Reviewer XMpz).
>
> **4. Line 191: “a sequence of image” vs. “a sequence of image tokens”**
>
> You are correct; the text should read “a sequence of image patches (tokens).”
>
> **5. How are object positions fixed in the images (line 320‑left)?**
>
> Object positions are sampled uniformly at random across the image canvas. There are no fixed regions or constraints; positions vary freely across images.
>
> Note that, to more directly test the role of spatial variance, we ran an additional experiment during the rebuttal period where object positions were fixed across images, thereby limiting translation invariance at the task level.
>
> Results were mixed. Across four random seeds, two models retained predominantly positional binding after visual training, supporting the role of translation invariance in driving the binding shift. In the other two seeds, however, the model still transitioned to symbolic binding despite the absence of spatial variability.
>
> We believe this is consistent with the use of pretrained image encoders, which likely encode translation‑invariant features due to their architecture and pretraining. Overall, this suggests that translation invariance is an important contributing factor, but not the sole mechanism underlying the observed binding shift. We will include these results and expand the discussion in the revised manuscript.
>
> **6. How is the “lightweight classifier” (Appendix G) trained?**
>
> We use a standard linear probing setup to measure representational content without introducing additional modeling capacity.
>
> Architecture: a single linear layer, nn.Linear(d_model, num_classes).
>
> Training: frozen activations with an 80/20 train–validation split, trained using cross‑entropy loss and Adam (lr = 0.01) for 100 epochs.
>
> Rationale: training independent probes per layer allows us to identify when semantic attributes (e.g., color identity) become linearly separable in the residual stream.
>
> We will add these specific training details to the appendix for completeness.
>
> **Minor details.**
> We will address, clarify, and correct all the minor points raised by the reviewer in the revised version of the manuscript. Thank you for these careful observations.

---

> > ### Author Rebuttal · Reviewer_YuhG · 2026-04-02
> >
> > I thank the authors for their feedback. It clearly addresses my concern. Accordingly, I confirm that this paper reflect good and inspiring research and I will raise my "score".

---

### Official Review · Reviewer_XMpz · 2026-03-10

**Soundness:** 2
**Presentation:** 3
**Significance:** 3
**Originality:** 2
**Overall Recommendation:** 4
**Confidence:** 3

**Summary:**

This paper investigates why VLMs sometimes outperform their base LLMs on purely text-only long-context retrieval tasks. Through controlled synthetic experiments and mechanistic interpretability the authors show that text-only training encourages a brittle positional binding shortcut. Subsequent training on visually rendered versions of the same task disrupts positional shortcuts through spatial translation invariance. This forces the model to adopt a more robust symbolic binding mechanism that persists even when text-only examples are reintroduced. The paper validates this mechanism shift both in small models trained from scratch and in pretrained Qwen LLM/VLM pairs. It also characterizes multiple circuit variants that implement symbolic binding.

**Compliance With Llm Reviewing Policy:**

Affirmed.

**Final Justification:**

My main concern on translation-invariance hypothesis is well-explained and I appreciate the authors for the additional experiments. I have updated the score accordingly.

**Key Questions For Authors:**

1. Can you directly test the translation-invariance hypothesis by manipulating the encoder's spatial properties? For example, comparing ViT with and without positional embeddings or randomizing patch order would isolate this factor.
2. How closely matched are the Qwen LLM and VLM variants in terms of instruction tuning and RLHF data? Could differences beyond visual training account for some of the observed gains?
3. Could you extend the interchange intervention analysis to report layer-wise distributions and per-head mechanisms? Connecting these to RoPE frequency bands as in [3] would directly test whether visual training reallocates heads toward symbolic-friendly frequencies.
4. Do the induced symbolic mechanisms transfer to natural long-context tasks such as subsets of LongBench or MDQA? Even limited pilot results would strengthen external validity.

**Limitations:**

The authors acknowledge the synthetic nature of the evaluation and the focus on the Qwen family. A more explicit discussion of how training mixture confounds might affect the large-model comparisons would be helpful.

**Strengths And Weaknesses:**

**Strengths**:
1. The central claim is conceptually important build on the traditional shortcut learning. Cross-modal vision training can causally shift a model's internal binding mechanism from positional to symbolic. This shift improves text-only generalization. The paper connects two active research threads on multimodal training effects and positional versus symbolic behavior in Transformers [1,2].
2. The controlled experimental pipeline includes text-only training, visual-only continuation with frozen encoders, and mixed fine-tuning. The underlying task is held fixed across modalities. The noise-token augmentation ablation is an informative control that rules out longer positional range as the main reason.
3. The mechanistic analysis is thorough. It uses interchange interventions, attention knockouts, and linear probes. Together these provide a concrete causal story connecting training data modality to internal mechanisms and downstream generalization.
4. Multiple visual encoders (ResNet, ViT, DINOv3) and multiple random seeds are explored. The effects persist across all these variations. The characterization of distinct symbolic circuit variants (color-key versus shape-key) goes beyond prior categorical labels of binding mechanisms.

**Weaknesses**:
1. The translation-invariance hypothesis is plausible but somewhat oversimplified. ViT-based encoders still inject positional embeddings and retain spatial structure. A more direct test would help, such as comparing ViT with and without positional embeddings or randomizing patch order and measuring the resulting binding mechanism. Without this, the causal role of translation invariance specifically remains only partially established.
2. The large-model validation is limited to the Qwen family. The LLM and VLM variants may differ in instruction tuning data, RLHF procedures, or other training mixture details beyond visual training alone. Including at least one additional model family (such as Llama or Gemma) and controlling for training mixture differences would strengthen the generality claim.
3. All tasks and evaluation benchmarks are synthetic on simple symbols. There are no transfer experiments to natural long-context settings such as LongBench variants or MDQA-like datasets. Even limited pilot results on such benchmarks would better establish external validity of the mechanism shift.
4. The symbolic-to-positional ratio for large models is reported at a single peak layer. Reporting distributions across layers or head-wise aggregates would help assess the stability and localization of the effect. Some methodological details of the interchange interventions could also be made more explicit in the main text rather than being deferred entirely to the appendix.

[1] How Do Language Models Bind Entities in Context?, ICLR 2024. \
[2] Mixing Mechanisms: How Language Models Retrieve Bound Entities In-Context, ICLR 2026.

---

> ### Author Rebuttal · Authors · 2026-03-31
>
> We thank the reviewer for the detailed and insightful feedback, as well as for recognizing the conceptual importance of our results and the rigor of our experimental and mechanistic analyses. Below, we address the main concerns point by point.
>
> **1. Translation invariance hypothesis**
>
> We partially agree. Our central contribution is to identify and explain a robust phenomenon: VLMs systematically outperform their LLM counterparts on text‑only retrieval tasks, and this improvement corresponds to a shift from positional to symbolic binding.
>
> An open question is why visual training induces symbolic binding. Our working hypothesis is that translation invariance plays an important role by weakening position‑based shortcuts, but we acknowledge that other factors may also contribute.
>
> To probe this more directly, during the rebuttal period we conducted an additional controlled experiment that limited translation invariance by fixing object positions across all images. Across four random seeds, results were mixed: in two cases, the model retained positional binding after visual training (supporting the role of translation invariance), while in the other two cases it still transitioned to symbolic binding.
>
> We believe these results align with the reviewer’s observation that pretrained image encoders incorporate multiple inductive biases—translation invariance among them—arising from architecture and pretraining. Even when spatial variability is reduced at the task level, other encoder‑level biases may still discourage positional binding. We will include these results and clarify in the revised paper that translation invariance is an important contributing factor, but not necessarily the only one.
>
> **2. How closely matched are the Qwen LLM and VLM variants in terms of instruction tuning and RLHF data? Could differences beyond visual training account for some of the observed gains?**
>
> It is possible that differences in instruction tuning or RLHF data contribute to the observed gains. Unfortunately, these datasets are not publicly documented for Qwen, making full isolation difficult.
>
> This limitation motivated our controlled synthetic setting, where we show that adding only visual‑modality training—without instruction tuning or RLHF—induces both a binding‑strategy shift and large OOD improvements. Thus, instruction tuning is not required for the emergence of symbolic binding.
>
> To test generality, we replicated our findings using an independent model family: InternLM‑3 (LLM) and InternVL‑3 (VLM). Despite different architectures and training pipelines, InternVL‑3 exhibits the same improvements in text‑only retrieval and the same shift toward symbolic binding. This cross‑family consistency suggests the effect arises from multimodal alignment rather than family‑specific data curation. We will add these results to the revised paper.
>
> Due to space constraints, we cannot add the data to the rebuttal directly, but we will update the paper's graphs with these new results. See [https://anonymous.4open.science/r/XMpz-P2-310C](https://anonymous.4open.science/r/XMpz-P2-310C).
>
> **3. Layer‑wise distributions and stability**
>
> We agree that relying on a single peak layer is insufficient. To address this, we introduce a global binding score that averages the symbolic‑to‑positional ratio across the full binding window rather than a single layer.
>
> This aggregated metric confirms a consistent shift toward symbolic binding across all models tested (Qwen‑2/2.5/3 and InternLM‑3). We will include full layer‑wise distributions and the new global metrics in the revised paper:
>
> Qwen 2 LLM=1.481 VLM=1.609\
> Qwen 2.5 LLM=1.589 VLM=1.846\
> Qwen 3 LLM=1.112 VLM=1.218\
> InternLM 3 LLM=1.676 VLM=2.044\
> (global ratio LLM vs VLM)
>
> Due to space constraints, we cannot include the per-layer data directly in text. For the sake of completeness, we have uploaded the revised figures. The first is the per-layer symbolic-to-positional ratio. The second figure corresponds to the same ratio, but this time quantified to a more representative measure. This is done by designating a set of binding layers for each model and taking the average symbolic-to-positional ratio. See [https://anonymous.4open.science/r/XMpz-P3-902Bj](https://anonymous.4open.science/r/XMpz-P3-902B).
>
> **4. Transfer to natural long‑context benchmarks**
>
> We agree that evaluating transfer to natural long‑context benchmarks (e.g., LongBench or MDQA) is an important direction. However, our current framework uses a minimal vocabulary and architecture by design to isolate binding mechanisms as cleanly as possible. Introducing full natural language would confound this analysis with additional variability.
>
> We view our synthetic setting as a necessary foundation for mechanistic understanding. Extending this analysis to natural benchmarks while preserving mechanistic clarity is a valuable but non‑trivial direction for future work, which we will explicitly discuss as a limitation in the final paper.

---

> > ### Author Rebuttal · Reviewer_XMpz · 2026-04-02
> >
> > My main concern on translation-invariance hypothesis is well-explained and I appreciate the authors for the additional experiments. I have updated the score accordingly.

---

### Official Review · Reviewer_3dB7 · 2026-03-11

**Soundness:** 3
**Presentation:** 3
**Significance:** 2
**Originality:** 3
**Overall Recommendation:** 4
**Confidence:** 3

**Summary:**

Through a combination of controlled synthetic experiments and mechanistic interpretability on the Qwen model family, the authors demonstrate that visual training forces a transition from brittle "positional shortcuts" to robust "symbolic binding" mechanisms.

**Compliance With Llm Reviewing Policy:**

Affirmed.

**Final Justification:**

I appreciate the authors' thorough rebuttal and acknowledge that the experimental work is technically solid and well-executed within its scope. While the analysis is rigorous for synthetic primitives, I remain skeptical about whether these findings can be meaningfully extrapolated to the complexities of real-world reasoning.

**Key Questions For Authors:**

See in the weakness.

**Limitations:**

Yes

**Strengths And Weaknesses:**

## Strength

1. The paper provides a compelling causal explanation for why multimodal training improves unimodal performance, moving beyond surface-level metrics to identify specific internal "binding circuits".

2. The study convincingly shows that the performance gains are not merely a result of exposure to longer sequences (via patch tokens) but are due to a qualitative shift in computational strategy that persists even when the model returns to text-only tasks.

## Weaknesses
1. The evaluation is primarily focused on synthetic and direct/indirect retrieval tasks.
2. The study defines "binding" through highly simplified synthetic primitives, such as color-shape associations. While this facilitates mechanistic analysis, to what extent can we view this specific form of "symbolic binding" as a representative proxy for the broader, more abstract reasoning required in natural language understanding? Is the identified "positional shortcut" merely an artifact of the synthetic task's rigidity?
3. The findings explain why VLMs exhibit superior text-only retrieval performance. Beyond this explanatory value, how should these insights inform the next generation of model architectures or training recipes?

---

> ### Author Rebuttal · Authors · 2026-03-31
>
> We thank the reviewer for their thoughtful and constructive feedback, and for recognizing several key contributions of our work. In particular, we appreciate the acknowledgement that our study provides a causal and mechanistic explanation for the benefits of multimodal training, identifying internal binding circuits. We also appreciate the recognition that the observed gains are not simply due to longer input sequences, but instead reflect a qualitative shift from positional shortcuts to symbolic binding, which persists even in text-only settings.
>
> The reviewer raises one primary concern regarding the focus on synthetic evaluation settings, and also highlights two important questions concerning the relationship between symbolic binding and broader reasoning, as well as how our findings can inform future architectures and training recipes. We address the evaluation concern directly and discuss the latter points to clarify the scope and implications of our work.
>
> **Focus on synthetic evaluation**
>
> We thank the reviewer for this important point and agree that evaluating on more complex, real-world tasks is a valuable direction. Our current focus on synthetic datasets is deliberate and follows prior work on binding (e.g., Feng & Steinhardt, 2024; Gur-Arieh et al., 2025), where controlled environments enable systematic analysis under well-defined conditions, precise interventions, and rigorous OOD evaluation.
>
> In our setting, synthetic data allows us to isolate the mechanisms of interest while minimizing confounding factors such as natural language variability or ambiguous inputs, which can otherwise make it difficult to attribute observed behaviors to specific internal processes. This level of control is particularly important for mechanistic interpretability, where clarity of analysis is a primary goal.
>
> Regarding the use of direct and indirect retrieval tasks, these provide a simple and well-established framework to study binding—the association between entities and their attributes—which is central to our work. We see this as a minimal testbed that enables clear inspection of model behavior and facilitates comparison with prior studies.
>
> We fully agree that extending these findings to more naturalistic settings is an important next step. We view our current evaluation as a controlled foundation that can support such future investigations, and we will clarify this motivation and limitation more explicitly in the revised version.
>
> **Relationship between symbolic binding and broader reasoning**
>
> We do not view the form of binding studied here as a direct proxy for broader abstract reasoning; rather, we see it as a necessary—but not sufficient—capability. Any model capable of more advanced reasoning must first be able to reliably bind attributes to entities, i.e., assign and maintain consistent meaning to its internal representations. (Feng & Steinhardt, 2024)
>
> At the same time, we agree that understanding how such mechanisms extend to richer forms of reasoning is an important open question. We view this as a promising direction for future work, particularly in developing evaluation settings that capture more complex reasoning while preserving interpretability.
>
> Regarding the positional shortcut identified in our setting, while our results provide evidence of this behavior, we agree that further investigation is needed to assess its generality beyond the current synthetic setup. Evaluating this phenomenon across a broader range of tasks and domains would help clarify its robustness and scope. We will incorporate this discussion in the revision to better contextualize the implications of our findings.
>
> **Implications for future architectures and training**
>
> We thank the reviewer for this insightful question. Our findings suggest that multimodal training may serve not only to improve performance within each modality, but also to encourage the emergence of more robust internal mechanisms that support out-of-distribution generalization.
>
> From this perspective, one implication is that multimodal training could be viewed not merely to expand capability coverage, but also as a tool for shaping the underlying computational strategies learned by the model. This opens the door to training approaches that explicitly leverage multimodal signals to promote more robust and generalizable representations, even when the target deployment setting is unimodal.
>
> We agree that further work is needed to translate these insights into concrete architectural or training design choices. We will expand this discussion in the final version to better highlight the potential implications and future directions suggested by our results.

---

> > ### Author Rebuttal · Reviewer_3dB7 · 2026-04-01
> >
> > I appreciate the authors' thorough rebuttal and acknowledge that the experimental work is technically solid and well-executed within its scope.
> >
> > I would like to clarify that I am not a specialist in the specific sub-field of mechanistic binding. However, from a broader perspective, my primary concern remains the general significance and representative value of this research. While the analysis is rigorous for synthetic primitives, I remain skeptical about whether these findings can be meaningfully extrapolated to the complexities of real-world reasoning in large-scale models.

---

> > > ### Author Response · Authors · 2026-04-08
> > >
> > > > **Reviewer comment**:
> > > >
> > > > “While the analysis is rigorous for synthetic primitives, I remain skeptical about whether these findings can be meaningfully extrapolated to the complexities of real‑world reasoning in large‑scale models.” (poner en quote en open review)
> > >
> > > We appreciate this concern and agree that extrapolation beyond controlled settings must be carefully justified. Our study deliberately focuses on information‑retrieval–style tasks because prior work has shown that mechanistic interventions for probing internal bindings in LLMs are feasible in relatively structured contexts (Gur-Arieh et al., 2025). For this reason, even though we evaluate relatively large models, our experiments necessarily “force” them into somewhat over‑structured information retrieval settings, where such interventions can be meaningfully applied.
> > >
> > > That said, we believe there are principled reasons to expect our findings to generalize to real‑world reasoning tasks. Most (if not all) reasoning problems rely on in‑context learning, where key information required for solving the task is present in the prompt rather than encoded in the model’s parameters. Consequently, successful reasoning requires the ability to reliably retrieve and manipulate relevant information from the context.
> > >
> > > To make this concrete, consider the classic example from the Chain‑of‑Thought (CoT) literature (Wei et al., 2022):
> > >
> > > ```
> > > **Context**:
> > > Q: Roger has 5 tennis balls. He buys 2 more cans of tennis balls. Each can has 3 tennis balls. How many tennis balls does he have now?
> > > A: Roger started with 5 balls. 2 cans of 3 tennis balls each is 6 tennis balls. 5 + 6 = 11. The answer is 11.
> > > Q: The cafeteria had 23 apples. If they used 20 to make lunch and bought 6 more, how many apples do they have?
> > >
> > > **Model output**:
> > > The cafeteria had 23 apples originally. They used 20 to make lunch. So they had 23 − 20 = 3. They bought 6 more apples, so they have 3 + 6 = 9. The answer is 9.
> > > ```
> > >
> > > Solving this problem requires correctly retrieving and integrating multiple pieces of information from the context (“23 apples,” “used 20,” “bought 6”), in the appropriate order. Failures in information retrieval—e.g., missing or misbinding one of these elements—would immediately derail the reasoning process, regardless of the model’s arithmetic competence.
> > >
> > > Our intent in presenting this example is not to claim that information retrieval alone constitutes reasoning, but rather to emphasize that robust contextual retrieval is a necessary (though not sufficient) condition for reasoning. From this perspective, models that exhibit systematically stronger retrieval abilities over long contexts are likely to enjoy a tangible advantage in downstream reasoning tasks, including those encountered in more realistic settings.
> > >
> > > We will make this connection more explicit in the revised paper to clarify why controlled information‑retrieval experiments are not merely synthetic artifacts, but instead probe a core capability underlying real‑world reasoning.
> > >
> > > We agree, however, that this argument remains intuitive rather than definitive. Establishing whether—and how—our findings extend to more complex, less structured reasoning tasks is an important direction for future work. In the meantime, we believe our contribution is threefold: (i) shedding light on the surprising phenomenon that VLMs can outperform their LLM counterparts on certain information retrieval tasks, (ii) grounding this observation mechanistically through the study of bindings, and (iii) marking an early milestone in understanding how different training modalities shape these foundational cognitive capacities.
> > >
> > >
> > > **Reference**
> > >
> > > Wei et al. (2022). Chain‑of‑thought prompting elicits reasoning in large language models. NeurIPS 35, 24824–24837.
> > >
> > > Gur-Arieh et al. (2025). Mixing Mechanisms: How Language Models Retrieve Bound Entities In-Context. arXiv preprint arXiv:2510.06182.

---

### Official Review · Reviewer_NZCB · 2026-03-12

**Soundness:** 3
**Presentation:** 3
**Significance:** 3
**Originality:** 3
**Overall Recommendation:** 5
**Confidence:** 3

**Summary:**

This paper explores why VLMs often outperform their text-only counterparts on textual tasks. Through controlled experiments and mechanistic interpretability, the authors demonstrate that text-only models rely on brittle "positional shortcuts" that are easy to fail in OOD scenarios. In contrast, visual training—due to its inherent translation invariance—forces the model to adopt a more robust "symbolic binding" strategy, where it explicitly links semantic attributes (like color or shape) rather than just memorizing token positions.
Overall, this work explores an important concept regarding how multimodal training acts as a powerful inductive bias to improve reasoning, and this research's broad aspect concerns the intersection of mechanistic interpretability and model robustness.

**Compliance With Llm Reviewing Policy:**

Affirmed.

**Key Questions For Authors:**

No question

**Limitations:**

The authors do not discuss the limitation, see Weaknesses.

**Strengths And Weaknesses:**

**Strengths**
1. The motivation is strong and the experimental setup is reasonable including priliminary experiments as well as the interpretation experiments.
2. The findings that LLM based more on "position binding" is interesting.


**Weaknesses**
1. The authors attribute the shift from positional to symbolic binding primarily to the nature of visual data (translation invariance). However, this conclusion may be premature, as it fails to adequately account for the influence of the pre-trained vision encoder (e.g., ViT). Since these encoders are often trained using contrastive learning, they may inherently inject specialized features or inductive biases into the language model that are distinct from the visual modality itself, potentially confounding the source of the improved reasoning strategy.
2. The experiments limits on information retrieval task, it is unclear whether the finding can be generalized to broader downstream tasks.

---

> ### Author Rebuttal · Authors · 2026-03-31
>
> Thank you for the encouraging and thoughtful review. We agree with the reviewer that this work explores an important mechanism by which multimodal training can act as an inductive bias that improves reasoning. Below, we address the two weaknesses raised.
>
> **1. Is translation invariance truly responsible for the shift from positional to symbolic binding?**
>
> We agree with the reviewer that attributing the shift in binding strategy solely to translation invariance would be premature, and we will clarify this point more carefully in the revised paper.
>
> Our results provide strong evidence that a systematic change in binding strategy occurs when transitioning from an LLM to a VLM, and our working hypothesis is that translation invariance plays a major role in this shift by disrupting position-based shortcuts. However, we do not claim that translation invariance is the only contributing factor.
>
> To investigate this question further, we ran an additional controlled experiment in which we explicitly limited translation invariance in the image modality by placing objects at fixed and consistent spatial locations across all images. Under this setup, we expected positional binding to remain effective. The results were mixed: in two out of four random seeds, the model indeed retained a predominantly positional binding strategy, supporting the role of translation invariance in driving the shift toward symbolic binding. However, in the remaining two seeds, the model still transitioned to symbolic binding despite the lack of spatial variability in the training examples.
>
> We believe these mixed outcomes are consistent with the reviewer’s suggestion. In particular, the pre-trained image encoder already incorporates multiple inductive biases—including, but not limited to, translation invariance—stemming from its architecture and pretraining objective. As a result, even when explicit spatial variation is reduced in our task, other encoder-level biases may still interfere with positional shortcuts learned during text-only training.
>
> We will include these additional results in the revised manuscript and expand the discussion to acknowledge that the observed binding shift likely arises from an interaction between visual data, encoder inductive biases, and multimodal training, rather than from translation invariance alone. We view fully disentangling these factors as an important and nontrivial direction for future work.
>
>
>
> **2. Limited evaluation to information retrieval tasks**
>
> We agree with the reviewer that it is important to understand whether these findings generalize beyond information retrieval tasks.
>
> We focus on information retrieval because it provides a clean and well-understood testbed for studying binding mechanisms, and because it allows us to apply established mechanistic interpretability tools (e.g., Gur‑Arieh et al., 2025) to reliably distinguish positional from symbolic binding. This choice enables us to isolate the binding mechanism itself in a controlled setting, rather than conflating it with other forms of reasoning.
>
> Importantly, binding entities to attributes for later retrieval is a fundamental component of in‑context learning and reasoning, and failures of long‑context generalization often reduce to failures of binding. In this sense, retrieval tasks serve as a diagnostic proxy rather than a narrow application domain. Our experimental setup follows prior work that uses retrieval-style tasks specifically to probe binding behavior (e.g., Feng & Steinhardt, 2024; Gur‑Arieh et al., 2025).
>
> That said, we agree that extending this analysis to broader downstream tasks is an important next step. We view our results as part of a first generation of efforts aimed at understanding how multimodal training reshapes internal reasoning strategies, and we will clarify this positioning more explicitly in the paper.

---

> > ### Author Rebuttal · Reviewer_NZCB · 2026-04-04
> >
> > Thanks for the rebuttal, and I would like to see the additional results in the updated version, I decide to keep my positive score.

---

### Decision · Program_Chairs · 2026-04-30

**Decision:**

Accept (regular)

**Comment:**

The paper investigates the surprising result that VLMs may sometimes outperform pure LLMs on text-only retrieval tasks. The study presents a series of experiments + mech interp analysis to show that text-only training yields positional binding shortcuts while subsequent training on visually-rendered versions of the same task drives the model towards a more robust symbolic binding mechanism.

On the positive side, reviewers commented on the originality of the contribution, the clarity of the results, the rigor of the experiments + the reproducibility of the work.

On the negative side, reviewers raised some concerns about the role of pretrained vision encoder features, the limited scope and the restricted validation limited to the Qwen model family. The rebuttal included experiments to test the translation-invariance hypothesis and clarified the positioning. Post rebuttal, all reviewers converged on accept with 3 reviewers raising their score. While 1 reviewer maintained some skepticism as to whether findings based on synthetic symbolic primitives will transfer to natural long-context reasoning (this is acknowledged by the authors). The AC flags this as important future work but not a blocker for acceptance as the mechanistic contribution stands on its own.

Overall the AC recommends the paper to be accepted